# Learning from Aggregate responses: Instance Level versus Bag Level Loss Functions

**Adel Javanmard[1,2], Lin Chen[1], Vahab Mirrokni[1], Ashwinkumar Badanidiyuru[1], Gang Fu[1]**
[1]Google Research, [2]University of Southern California
{adeljavanmard,linche,mirrokni,ashwinkumarbv,thomasfu}@google.com

## Abstract

Due to the rise of privacy concerns, in many practical applications, the training data is aggregated before being shared with the learner to protect the privacy of users' sensitive responses. In an aggregate learning framework, the dataset is grouped into bags of samples, where each bag is available only with an aggregate response, providing a summary of individuals' responses in that bag. In this paper, we study two natural loss functions for learning from aggregate responses: the bag-level loss and the instance-level loss. In the former, the model is learned by minimizing a loss between the aggregate responses and aggregate model predictions, while in the latter, the model aims to fit individual predictions to the aggregate responses. In this work, we show that the instance-level loss can be perceived as a regularized form of the bag-level loss. This observation allows us to compare the two approaches with respect to the bias and variance of the resulting estimators and to introduce a novel interpolating estimator that combines the two approaches. For linear regression tasks, we provide a precise characterization of the risk of the interpolating estimator in an asymptotic regime where the size of the training set grows in proportion to the feature dimension. Our analysis enables us to theoretically understand the effect of different factors, such as bag size, on the model's prediction risk. Additionally, we propose a mechanism for differentially private learning from aggregate responses and derive the optimal bag size in terms of the prediction risk-privacy trade-off. We also carry out thorough experiments to corroborate our theory and show the efficacy of the interpolating estimator.

## 1 Introduction

Machine learning has revolutionized many industries and aspects of our lives, but its widespread use has also raised concerns about privacy. One way to address these concerns is to use aggregate labels, which are labels assigned to groups of data points rather than to individual data points (Criteo Privacy Preserving ML Competition, 2021). Since even before the machine learning era, aggregate labels have been commonly used in group testing, a method that combines samples from various individuals or objects to maximize the use of limited resources (Wein & Zenios, 1996; Sunjaya & Sunjaya, 2020). For example, group testing is used to screen for HIV in donated blood products and to identify viral epidemics, such as COVID-19. In recent years, there has been growing interest in using aggregate labels to train machine learning models (Papernot et al., 2016; Al-Rubaie & Chang, 2019; De Cristofaro, 2020). This approach has the potential to preserve privacy while still allowing for the development of accurate and effective models. The SKAdNetwork API from Apple is an example of how aggregate labels can be used to preserve privacy in machine learning (Apple Developer Documentation, 2023). SKAdNetwork provides advertisers with insights into the effectiveness of their ad campaigns without compromising the privacy of users. This is achieved by using aggregate labels to represent groups of users who have interacted with an ad. For example, an advertiser might receive a report indicating that 10% of users who saw their ad installed their app. However, the advertiser would not be able to see the specific identities of those users. Another example is the Private Aggregation API of Chrome Privacy Sandbox. This API collects instance-label pairs from users, but it protects their privacy by providing apps and services with bags of instances that are labeled in an aggregated way. The aggregate label can be further perturbed to ensure differential privacy, which is a mathematical guarantee that the data cannot be used to identify individuals.

To formally describe the setup, consider a dataset consisting of $n$ samples $(\boldsymbol{x}_i, y_i)$, for $i \in [n]$, with $\boldsymbol{x}_i \in \mathbb{R}^d$ being the features vector and $y \in \mathbb{R}$ being the response variable. Motivated by applications where the response variables $y_i$ carry private information, instead of sharing individual responses with the learner, only aggregate responses are shared as follows: A set of $m$ non-overlapping bags $B_a \subseteq [n]$, for $a \in [m]$, is formed, each of size $k$. For each bag, the average response $\bar{y}_a = (\sum_{i \in B_a} y_i)/k$ is shared with the learner, along with the individual features vectors $\boldsymbol{x}_i$.

There are two common choices for learning from aggregated data:

- **Bag-level loss:** The model is learned by minimizing a loss function that measures the distance between aggregate responses and the aggregate model predictions. Namely, $\widehat{\boldsymbol{\theta}} = \arg\min_{\boldsymbol{\theta}} \mathcal{L}_{\text{bag}}(\boldsymbol{\theta})$ with

$$\mathcal{L}_{\text{bag}}(\boldsymbol{\theta}) = \frac{1}{m} \sum_{a=1}^{m} \ell\Big(\bar{y}_a, \frac{1}{k} \sum_{i \in B_a} f_{\boldsymbol{\theta}}(\boldsymbol{x}_i)\Big), \tag{1}$$

where $\ell(\cdot, \cdot) : \mathbb{R} \times \mathbb{R} \to \mathbb{R}_{\geq 0}$, and $f_{\boldsymbol{\theta}}$ is a family of models parameterized by $\boldsymbol{\theta}$.

- **Instance-level loss:** The model is learned by minimizing the loss between the aggregate responses and individual model predictions. Namely, $\widehat{\boldsymbol{\theta}} = \arg\min_{\boldsymbol{\theta}} \mathcal{L}_{\text{ins}}$ with

$$\mathcal{L}_{\text{ins}}(\boldsymbol{\theta}) := \frac{1}{mk} \sum_{a=1}^{m} \sum_{i \in B_a} \ell(\bar{y}_a, f_{\boldsymbol{\theta}}(\boldsymbol{x}_i)). \tag{2}$$

An advantage of bag-level loss is that since it involves aggregate model predictions, it provides a layer of protection for the features, in addition to responses. For example, in the case of linear models, the learner can still minimize the bag-level loss, given only access to the aggregate responses and the aggregate features.

Despite the common use of bag-level loss and instance-level loss, we still lack a clear understanding of the performance of models learned by each approach, as measured in terms of model generalization error. One of our contributions in this paper is to establish a connection between the two losses and shed light on when one outperforms the other in terms of model generalization. We build a strong intuitive understanding through precise quantitative statements. The key observation is that employing a bag-level loss leads to models with reduced bias but increased variance, as opposed to models trained with instance-level loss. Therefore, in scenarios with diverse responses, instance-level loss is more effective. Conversely, for use cases with more homogeneous responses, the bag-level loss is preferable, as it proves more effective in reducing bias.

## 1.1 SUMMARY OF CONTRIBUTIONS AND ORGANIZATION OF THE PAPER

Our contributions are summarized as follows:

$(i)$ We show that the instance-level loss can be perceived as the bag-level loss with an additive regularization term. While the regularization penalty introduces bias into the estimator, it also serves to reduce variance when the sample size $(n)$ is comparable to the feature dimension $(d)$. Depending on the interplay between bias and variance, one loss may outweigh the other, resulting in a model with better generalization. Motivated by this observation, we propose a new estimator that includes a tuning parameter $\rho \in [0, 1]$ to control the strength of the regularization and, hence, the bias-variance trade-off. The case of $\rho = 0$ corresponds to the bag-level loss, and the case of $\rho = 1$ corresponds to the instance-level loss. However, by optimally tuning $\rho$, we can obtain models that outperform both of these cases.

$(ii)$ We next focus on the so-called *proportional regime* where the sample size $(n)$ and the feature dimension $(d)$ are of the same order, i.e., $n/d \to \psi$, for some arbitrary bounded constant $\psi \in (0, \infty)$, as $n \to \infty$. This asymptotic regime has attracted significant interest in recent years due to its relevance in practice, where the classical population regime $(n/d \to \infty)$ fails to capture the behavior of overparametrized machine learning models (see e.g., Hastie et al. (2022); Javanmard et al. (2020); Hassani & Javanmard (2022); Mei & Montanari (2022)). In Section 2.2, we focus on linear models and derive a precise characterization of model generalization error for both the bag-level and the instance-level losses. Our precise theory captures the effect of different

quantities of interest, such as bag size $k$, overparametrization, signal-to-noise ratio of the data, and the regularization parameter $\rho$ on the model performance. It also allows us to compare the two approaches in terms of bias and variance and theoretically find the optimal regularization parameter $\rho$, as described in the previous item.

($iii$) In Section 4, we propose a differentially private mechanism for learning from aggregate data. It is based on the Laplace mechanism, which adds Laplace noise to the aggregate responses to ensure $\varepsilon$-(label) differential privacy. For a given privacy loss $\varepsilon$, our theory in Section 2.2 allows us to derive the optimal size of the bags that results in the best model generalization. Interestingly, the singleton bags are not always the optimal choice, which implies that applying the Laplace mechanism to aggregate data gives a better privacy-model generalization trade-off compared to directly applying the Laplace mechanism to individual responses.

## 1.2 RELATED WORK

There is a rich body of prior work on learning with label proportions (LLP) (Shi et al., 2018; 2019; Cui et al., 2017; Xiao et al., 2020; Li & Wang, 2018; Quadrianto et al., 2008; Patrini et al., 2014; Musicant et al., 2007; Zhang et al., 2020; Qi et al., 2017; 2016; Chen et al., 2017; Lu et al., 2019; Chen et al., 2023; Javanmard et al., 2024). In what follows, we categorize previous work into bag-level, instance-level and other methods.

**Bag-level methods.** Rueping (2010) proposed a large-margin support vector regression (SVR) method for learning with label proportions (LLP). The proposed approach models the mean of each bag as a "super-instance" with a soft label equal to the label proportion of the bag. Yu et al. (2014) introduced the Empirical Proportion Risk Minimization (EPRM) framework, which minimizes the bag-level loss function. Yu et al. (2014) also derived a VC-dimension-based uniform convergence bound for the gap between the empirical and population bag-level loss functions. Ardehaly & Culotta (2017) applied the aggregated cross-entropy loss to deep learning and classification problems.

**Instance-level methods.** Yu et al. (2013) proposed a method that also uses the idea of support vector regression (SVR), but models the probability of each instance rather than each bag, in contrast to Rueping (2010). The proposed loss function has two components: (1) the sum of the hinge losses between the unknown individual label and the predicted label of each instance, and (2) the sum of the losses between the label proportion of the unknown individual labels of each bag and the true label proportion of that bag. Dulac-Arnold et al. (2019) uses a similar idea to (Rueping, 2010) and considers relaxation to make the optimization problem more tractable. For classification problems, Busa-Fekete et al. (2023) derived an unbiased estimator of the individual labels of the data examples. This estimator is a function of the label proportion of the bag to which the example belongs and the probability distribution of all labels in the population. By using this estimator of the individual labels, one can apply the usual supervised learning methods.

**Other related works.** Quadrianto et al. (2008) proposed a kernelized conditional exponential model for inferring the individual labels of unseen examples based on training examples grouped in bags and the label proportion of the bags. The method is based on maximizing the log-likelihood of the model. A key assumption of the model is that the features of an example are conditionally independent of the bag to which it belongs, given the example's label. Fish & Reyzin (2017) formally defines the class of functions that can be learned from label proportions (LLP Learnable) and resolves foundational questions about the computational complexity of LLP and its relationship to PAC learning. Scott & Zhang (2020) solves the LLP problem through reduction to mutual contamination models (MCMs). Saket (2021) investigated the learnability of linear threshold functions (LTFs) from label proportions. Saket et al. (2022) proposed a method for combining bag distributions to improve learning from label proportions. While in this work we focus on random bagging, other bagging schemes based on features are proposed recently by Chen et al. (2023) and Javanmard et al. (2024).

## 2 MAIN RESULTS

### 2.1 INTUITION: INSTANCE-LEVEL LOSS ACTS AS A REGULARIZED BAG-LEVEL LOSS

In this work we establish a connection between bag-level and instance-level losses, showing that the latter can be perceived as a regularized version of the former. We first show this claim for quadratic loss and then discuss an extension to general convex losses.

**Lemma 2.1.** *Consider the quadratic loss $\ell(x, y) = (x - y)^2$. For the bag-level loss (1) and the instance-level loss (2) we have*

$$\mathcal{L}_{\text{ins}}(\boldsymbol{\theta}) = \mathcal{L}_{\text{bag}}(\boldsymbol{\theta}) + \mathcal{R}(\boldsymbol{\theta}),$$

*where the regularization term $\mathcal{R}(\boldsymbol{\theta})$ is given by*

$$\mathcal{R}(\boldsymbol{\theta}) := \frac{1}{k} \sum_{a=1}^{m} \sum_{i,j \in B_a} (f_{\boldsymbol{\theta}}(\boldsymbol{x}_i) - f_{\boldsymbol{\theta}}(\boldsymbol{x}_j))^2. \tag{3}$$

Note that the regularization term $\mathcal{R}(\boldsymbol{\theta})$ does not depend on the responses, but it does depend on the feature vectors. While traditional regularization techniques only depend on model parameters, there is also a growing body of work on data-dependent regularization. This type of regularization explicitly takes into account the training data during the regularization process, which can lead to improved generalization performance in certain settings (Shivaswamy & Jebara, 2010; Zhao et al., 2019; Mou et al., 2018). In addition, $\mathcal{R}(\boldsymbol{\theta})$ captures the within cluster variations of the model predictions. While the additive regularization induces bias to the estimator minimizing the instance-level loss, it can reduce the variance when the sample size and features dimension are comparable.

Also note that when the bag-level loss does not have a unique minimizer, the regularization term promotes solutions with smaller $\mathcal{R}(\boldsymbol{\theta})$. Therefore, if the true model also has small penalty value, this will help to impose this structure on the estimator.

Motivated by Lemma 2.1, we introduce an interpolating loss $\mathcal{L}_{\text{int}}(\boldsymbol{\theta})$ which keeps the same form of the regularization but includes a tuning parameter to control the strength of the regularization, namely

$$\mathcal{L}_{\text{int}}(\boldsymbol{\theta}) := \mathcal{L}_{\text{bag}}(\boldsymbol{\theta}) + \rho \mathcal{R}(\boldsymbol{\theta}), \quad \rho \in [0, 1]. \tag{4}$$

In practice, the optimal value of $\rho$ can be set via cross-validation to obtain the best model performance. In the next section, we provide a theory for linear models which also allows us to analytically derive the optimal choice of $\rho$.

We conclude this section by extending lemma 2.1 to other loss functions.

**Lemma 2.2.** *Consider the loss function $\ell : \mathbb{R} \times \mathbb{R} \to \mathbb{R}_{\geq 0}$ with continuous second derivative in the second argument. Also suppose that $|\frac{\partial^2}{\partial b^2} \ell(a, b)| \leq C$ for a constant $C$. We then have the following relation between the bag-level loss (1) and instance-level loss (2):*

$$\mathcal{L}_{\text{ins}}(\boldsymbol{\theta}) \leq \mathcal{L}_{\text{bag}}(\boldsymbol{\theta}) + C\mathcal{R}(\boldsymbol{\theta}), \tag{5}$$

*where the regularization $\mathcal{R}(\boldsymbol{\theta})$ is given by (3). In addition, if $\ell(\cdot, \cdot)$ is convex in the second argument we have $\mathcal{L}_{\text{bag}}(\boldsymbol{\theta}) \leq \mathcal{L}_{\text{ins}}(\boldsymbol{\theta})$ for any model $\boldsymbol{\theta}$.*

We refer to the supplementary for the proof of Lemmas 2.1 and 2.2.

## 2.2 LINEAR MODELS: A PRECISE THEORY

Consider a dataset of $n$ i.i.d pairs $(\boldsymbol{x}_i, y_i)$, for $i \in [n]$, with $\boldsymbol{x}_i \in \mathbb{R}^d$ a feature vector and $y_i \in \mathbb{R}$ a response variable. We assume the following linear model for the responses:

$$y_i = \boldsymbol{x}_i^T \boldsymbol{\theta}_0 + w_i,$$

where $w_i$ are i.i.d noise with $\mathbb{E}(w_i) = 0$ and $\text{Var}(w_i) = \sigma^2$. We collect the responses into a vector $\boldsymbol{y} \in \mathbb{R}^n$ and the features in a matrix $\boldsymbol{X} \in \mathbb{R}^{n \times d}$ (with rows $\boldsymbol{x}_i \in \mathbb{R}^d$).

We consider a setting where the features are i.i.d Gaussian vectors $\boldsymbol{x}_i \sim N(0, \boldsymbol{I}_d)$. Consider a model $\boldsymbol{\theta}$, trained on the data $\boldsymbol{X}, \boldsymbol{y}$ and a test point $\boldsymbol{x}_{\text{test}} \sim N(0, \boldsymbol{I}_d)$, independent of the training data $\boldsymbol{X}, \boldsymbol{y}$. The prediction risk of $\boldsymbol{\theta}$ is defined by

$$\text{Risk}_{\boldsymbol{X}}(\boldsymbol{\theta}) = \mathbb{E}[(\boldsymbol{x}_{\text{test}}^T \boldsymbol{\theta} - \boldsymbol{x}_{\text{test}}^T \boldsymbol{\theta}_0)^2 | \boldsymbol{X}] = \mathbb{E}[\|\boldsymbol{\theta} - \boldsymbol{\theta}_0\|^2 | \boldsymbol{X}].$$

Note that our definition of the risk is conditional on $\boldsymbol{X}$ and is made explicit in our notation $\text{Risk}_{\boldsymbol{X}}$. In addition, the bias-variance decomposition of the risk is given by

$$\text{Risk}_{\boldsymbol{X}}(\boldsymbol{\theta}) = \text{Bias}_{\boldsymbol{X}}(\boldsymbol{\theta}) + \text{Var}_{\boldsymbol{X}}(\boldsymbol{\theta}), \tag{6}$$

$$\text{Bias}_{\boldsymbol{X}}(\boldsymbol{\theta}) = \|\boldsymbol{\theta}_0 - \mathbb{E}[\boldsymbol{\theta}|\boldsymbol{X}]\|^2, \quad \text{Var}_{\boldsymbol{X}}(\boldsymbol{\theta}) = \text{tr}(\text{Cov}(\boldsymbol{\theta}|\boldsymbol{X})).$$

Recall our interpolating loss given by (4). Specializing to linear model $f_{\boldsymbol{\theta}}(\boldsymbol{x}) = \boldsymbol{x}^T \boldsymbol{\theta}$ and quadratic loss, we arrive at

$$
\begin{aligned}
\mathcal{L}_{\text{int}}(\boldsymbol{\theta}) &= \mathcal{L}_{\text{bag}}(\boldsymbol{\theta}) + \rho \mathcal{R}(\boldsymbol{\theta}) \\
&= \mathcal{L}_{\text{bag}}(\boldsymbol{\theta}) + \rho(\mathcal{L}_{\text{ins}}(\boldsymbol{\theta}) - \mathcal{L}_{\text{bag}}(\boldsymbol{\theta})) \\
&= \frac{1}{2mk} \sum_{a=1}^{m} \sum_{i \in B_a} \left( (1-\rho)(\bar{y}_a - \boldsymbol{x}_a^T \boldsymbol{\theta})^2 + \rho(\bar{y}_a - \boldsymbol{x}_i^T \boldsymbol{\theta})^2 \right) ,
\end{aligned}
\tag{7}
$$

with $y_a$ and $\boldsymbol{x}_a$ respectively denoting the average response and average of feature vectors in bag $a$. We also define $\widehat{\boldsymbol{\theta}}_{\text{int}} := \arg\min_{\boldsymbol{\theta}} \mathcal{L}_{\text{int}}(\boldsymbol{\theta})$.

In this section, we derive a precise characterization of the bias and variance of $\widehat{\boldsymbol{\theta}}_{\text{int}}$. Our theory provides a precise understanding on the role of different factors, such as bags size $k$, overparametrization, and signal-to-noise ratio of the data. It also allows us to theoretically derive optimal regularization parameter $\rho$.

We next describe the the asymptotic regime of interest and our assumptions.

**Assumption 2.3. (Asymptotic Setting)** We focus on the so-called 'proportional' regime where the sample size $n$ and the features dimension $d$ grow in proportion as $d \to \infty$. Formally for an arbitrary but fixed constant $\psi \in (1, \infty)$, we have $n/d \to \psi$. We further assume that the size of bags $(k)$ remains fixed as $d \to \infty$, and the model norm $\|\boldsymbol{\theta}_0\|$ converges. For the sake of normalization and without loss of generality, we assume $\lim_{d \to \infty} \|\boldsymbol{\theta}_0\| = 1$.

We next state our assumption on the bagging structure.

**Assumption 2.4.** We assume that the bagging configuration is independent from the training data $(\boldsymbol{X}, \boldsymbol{y})$. In addition, we assume that the bags are non-overlapping, so that each sample appears in exactly one bag.

**Theorem 2.5.** *Consider data generated according to linear model in an asymptotic regime described in Assumption 2.3. Also suppose that the bags are of size $k$ and are non-overlapping as described in Assumption 2.4. The bias and variance of $\widehat{\boldsymbol{\theta}}_{\text{int}}$ are characterized below:*

- **Bias**: *For $k > 1$ and $\rho > 0$, let $\alpha_*$ be the nonnegative fixed point of the following equation:*

$$
\rho + \frac{\psi}{k(1-\alpha_*)} - 1 = \frac{\psi}{k\alpha_*} \rho(k-1) .
$$

  *The bias of $\widehat{\boldsymbol{\theta}}_{\text{int}}$ converges in probability to*

$$
\mathsf{Bias}_{\boldsymbol{X}}(\widehat{\boldsymbol{\theta}}_{\text{int}}) \overset{(p)}{\to} \alpha_*^2 + \frac{\alpha_*^2}{\frac{(k-1)\psi}{k^2(1-\alpha_*)^2} - \left(\frac{\alpha_*}{1-\alpha_*}\right)^2 \frac{1}{k} - \frac{k-1}{k}} .
\tag{8}
$$

  *If $k = 1$ or $\rho = 0$, then $\widehat{\boldsymbol{\theta}}_{\text{int}}$ is unbiased.*

- **Variance**: *Let $(v_*, u_*)$ be the solution of the following system of equations:*

$$
\begin{cases}
\dfrac{\psi}{1+u} + \dfrac{\rho\psi(k-1)}{\rho+u} = k , \\[2ex]
\dfrac{\psi(1+v)}{(1+u)^2} + \dfrac{\rho^2\psi(k-1)}{(\rho+u)^2} = k .
\end{cases}
$$

  *We then have*

$$
\mathsf{Var}_{\boldsymbol{X}}(\widehat{\boldsymbol{\theta}}_{\text{int}}) \overset{(p)}{\to} \frac{\sigma^2}{v_*} .
\tag{9}
$$

## 3 DISCUSSION

### 3.1 COMPARISON BETWEEN BAG-LEVEL AND INSTANCE-LEVEL ESTIMATORS

We next specialize the result of Theorem 2.5 to $\rho = 0$ (corresponding to the bag-level loss) and $\rho = 1$ (corresponding to the instance-level loss). Note that the bag-level loss is the ordinary least

square estimator with $m = n/k$ samples and $d$ parameters and therefore is well-defined only for $n \geq kd$. In the asymptotic regime of Assumption 2.3, this implies that $\psi = \lim_{d\to\infty} n/d \geq k$.)

The following corollary characterizes the bias and variance of the bag-level and the instance-level estimators.

**Corollary 3.1.** *The followings hold:*

- **Bag-level loss:** *Suppose* $\psi \geq k$. *For the bag-level estimator* $\widehat{\boldsymbol{\theta}}_{\text{bag}} := \arg\min_{\boldsymbol{\theta}} \mathcal{L}_{\text{bag}}(\boldsymbol{\theta})$, *we have*

$$\text{Bias}_{\boldsymbol{X}}(\widehat{\boldsymbol{\theta}}_{\text{bag}}) \overset{(p)}{\to} 0, \quad \text{Var}_{\boldsymbol{X}}(\widehat{\boldsymbol{\theta}}_{\text{bag}}) \overset{(p)}{\to} \frac{\sigma^2}{\frac{\psi}{k} - 1}.$$

- **Instance-level loss**: *For the instance-level estimator* $\widehat{\boldsymbol{\theta}}_{\text{ins}} := \arg\min_{\boldsymbol{\theta}} \mathcal{L}_{\text{ins}}(\boldsymbol{\theta})$, *we have*

$$\text{Bias}_{\boldsymbol{X}}(\widehat{\boldsymbol{\theta}}_{\text{ins}}) \overset{(p)}{\to} \left(1 - \frac{1}{k}\right)\left(1 + \frac{2 - \psi}{k(\psi - 1)}\right), \quad \text{Var}_{\boldsymbol{X}}(\widehat{\boldsymbol{\theta}}_{\text{ins}}) \overset{(p)}{\to} \frac{\sigma^2}{k(\psi - 1)}.$$

As we see from the above characterization, while the bag-level loss is unbiased, it has higher variance compared to the instance-level loss. The dominance of one estimator over the other in terms of prediction risk (6) depends on the relative magnitudes of bias and variance.

In the next lemma, we provide a threshold on the model signal-to-noise ratio (SNR), defined as $\text{SNR} = \|\boldsymbol{\theta}_0\|^2/\sigma^2$, under which the bag-level estimator has the smaller risk and above it the instance-level loss has the smaller risk.

**Lemma 3.2.** *Suppose* $\psi \geq k > 1$. *Then,* $\text{Risk}_{\boldsymbol{X}}(\widehat{\boldsymbol{\theta}}_{\text{ins}}) \leq \text{Risk}_{\boldsymbol{X}}(\widehat{\boldsymbol{\theta}}_{\text{bag}})$ *if and only if*

$$\text{SNR} := \frac{\|\boldsymbol{\theta}_0\|^2}{\sigma^2} \leq \frac{(k+1)\psi - k}{(\psi - k)(\psi(1 - \frac{1}{k}) - 1 + \frac{2}{k})}.$$

*For $k = 1$, we have* $\widehat{\boldsymbol{\theta}}_{\text{ins}} = \widehat{\boldsymbol{\theta}}_{\text{bag}}$ *and so they have the same risk.*

Intuitively, when the SNR is low it means that the variance of the samples is large relative to the model strength. In this case, the instance-level loss which aims to decrease the variance, at the cost of increasing bias, obtains a lower prediction risk than the bag-level estimator.

## 3.2 ROLE OF DIFFERENT FACTORS ON MODEL RISK

Our theory allows to precisely characterize the effect of different factors, namely the bag size, SNR, overparametrization and the regularization parameter on the bias, variance and risk of the estimator. In Figure 1 we plot the theoretical curves derived in Theorem 2.5 versus $\rho$. As we see by increasing $\rho$ (stronger regularization), the bias increases and the variance decreases. The curves for the risk (rightmost panel) exhibits an optimal $\rho$ that balances the trade-off between bias and variance. The first row of plots shows the effect of SNR on the curves. As expected, higher SNR reduces the model variance, but has no effect on the bias, and so reduces the risk. The second row of plots shows the effect of bag size $k$. Larger $k$ induces more bias. For small $\rho$, larger $k$ increases the variance, but at large $\rho$, increasing $k$ reduces the variance. The last row of figures shows the effect of $\psi = n/d$. Larger $\psi$ decreases bias, variance and the risk since it corresponds to more samples relative to the number of parameters to be estimated.

## 4 DIFFERENTIALLY PRIVATE AGGREGATE LEARNING

As we discussed aggregate learning framework provides some layer of privacy protections by obfuscating the individual responses and only revealing aggregate responses in each bag. In particular, when the bags are non-overlapping and each are of size at least $k$, replacing individual responses with the aggregate ones offers $k$-anonymity, a privacy notion, stating that any record is indistinguishable from at least $k-1$ other records, in terms of response. One can also argue about the information leakage in aggregate learning through the notion of mutual information between individual and aggregate responses, conditional on the features vectors.

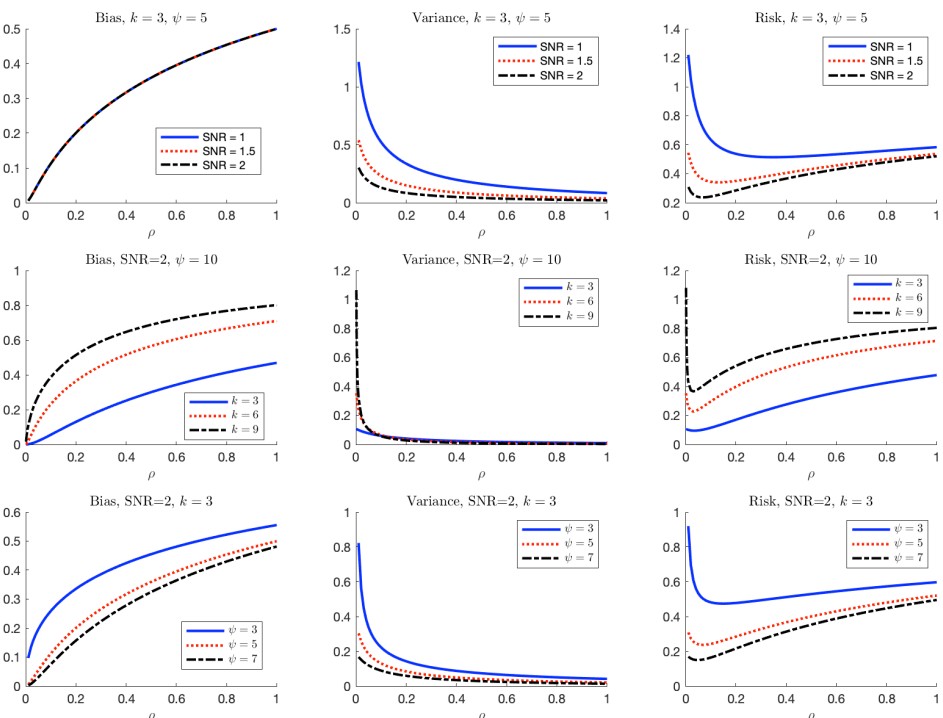

Figure 1: Effect of SNR, bag size ($k$) and overparametrization ($1/\psi = d/n$) on the bias, variance and the risk of the model. The curves are the theoretical curves given by Theorem 2.5.

Another widely used privacy notion is differential privacy (DP), put forward by Dwork et al. (2006b;a). Informally speaking, a mechanism (data processing algorithm) is called DP, if its output distribution does not change significantly if a single record is changed in the input dataset. We focus on the notion of label differential privacy, introduced by Chaudhuri & Hsu (2011), in which the changes only happen on the 'response' of a single record. Label DP is also widely used for applications where only the privacy of the responses are concerned, in opposite to all features. For example, in a medical test, the demographic features of patients may be far less private than the responses (medical test results). We recall the formal definition of label DP from (Chaudhuri & Hsu, 2011).

**Definition 4.1.** (Label Differential Privacy) Consider a randomized mechanism $\mathcal{M} : D \rightarrow \mathcal{O}$ that takes as input a dataset $D$ and outputs into $\mathcal{O}$. A mechanism $\mathcal{M}$ is called $\varepsilon$- label DP, if for any two datasets $(D, D')$ that differ in the label of a single example and any subset $O \subseteq \mathcal{O}$ we have

$$\mathbb{P}[\mathcal{M}(D) \in O] \leq e^\varepsilon \mathbb{P}[\mathcal{M}(D') \in O],$$

where $\varepsilon$ is the privacy budget.

It is easy to observe that the framework of learning from aggregate data, alone does not provide (label) differential privacy. In this section we propose a mechanism which adds noise to the aggregate (truncated) responses to ensure label DP guarantee. The procedure is outlined in Algorithm 1. The truncation level is of order $\sqrt{\log n}$, so that with high probability it does not happen. But this truncation step provides us an upper bound on the responses which we use to bound the sensitivity of aggregate responses and decide on the noise variance needed to ensure DP.

**Lemma 4.2.** *The mechanism described in Algorithm 1 is $\varepsilon$-label DP.*

With slight abuse of notation, we let $\widehat{\boldsymbol{\theta}}_{\text{int}}$ be the minimizer of the interpolating loss, when the learner uses the privatized aggregate responses $\tilde{y}_a$, $a \in [m]$ provided by Algorithm 1. By Lemma 4.2 and the post-processing property of differential privacy, $\widehat{\boldsymbol{\theta}}_{\text{int}}$ is also $\varepsilon$-label DP. In the next theorem, we use our theoretical result from Section 2.2 to characterize the risk of $\widehat{\boldsymbol{\theta}}_{\text{int}}$ in terms of privacy loss $\varepsilon$ and the bags size $k$, among other factors.

**Algorithm 1** Label differentially private learning from aggregate data

**Input:** individual responses $\{y_i\}_{i=1}^n$, bags $\{B_a\}_{a=1}^m$ each of size $k$, truncation level $C$.
**Output:** Privatized aggregate responses $\tilde{y}_1, \ldots, \tilde{y}_m$
1: *// Clip the responses*
2: **for** $i = 1, 2, \ldots, n$ **do**
3:   $y_i^c \leftarrow \max(\min(y_i, C\sqrt{\log n}), -C\sqrt{\log n})$
4: **end for**
5: **for** $a = 1, 2, \ldots, m$ **do**
6:   $\bar{y}_a \leftarrow \frac{1}{k} \sum_{i \in B_a} y_i$
7:   $\tilde{y}_a \leftarrow \bar{y}_a + z_a$ where $z_a \sim \text{Laplace}\left(\frac{C\sqrt{\log n}}{k\varepsilon}\right)$.
8: **end for**

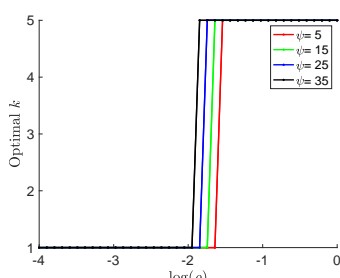

Figure 2: Optimal bag size $k$ as a function of $\log(\rho)$, for label DP learning from aggregate data. We observe a phase transition, where for some $\rho_*$ if $\rho < \rho_*$, $k = 1$ is the best choice while for $\rho > \rho_*$ the largest $k$ is optimal.

**Theorem 4.3.** *Let $\widehat{\boldsymbol{\theta}}_{\text{int}} \in \arg\min_{\boldsymbol{\theta}} \mathcal{L}_{\text{int}}(\boldsymbol{\theta})$ using the privatized aggregate responses from Algorithm 1, with $C^2 > 2(1 + \sigma^2)$. Let $(v_*, u_*)$ be the solution of the following system of equations:*

$$
\begin{cases}
\dfrac{\psi}{1+u} + \dfrac{\rho\psi(k-1)}{\rho+u} = k\,, \\[3mm]
\dfrac{\psi(1+v)}{(1+u)^2} + \dfrac{\rho^2\psi(k-1)}{(\rho+u)^2} = k\,.
\end{cases}
$$

*We then have*

$$
\frac{1}{\log n}\mathsf{Risk}_{\boldsymbol{X}}(\widehat{\boldsymbol{\theta}}_{\text{int}}) \overset{(p)}{\to} \frac{2C^2}{k\varepsilon^2}\frac{1}{v_*}\,. \tag{10}
$$

A virtue of Theorem 4.3 is that it allows to decide on the optimal bag size $k$, which minimizes the risk of the estimator $\widehat{\boldsymbol{\theta}}_{\text{int}}$. On the one side, large bag sizes allow the aggregate responses to be less sensitive to each individual response, which means smaller noise is needed to ensure $\varepsilon$-DP. On the other hand, at small $\rho$, larger $k$ can cause larger instability because it amounts to fewer bags and hence fewer (aggregate) labels. As $\rho$ increases, the regularization becomes stronger and brings more stability. In Figure 2 we plot optimal $k^*$ (from the bounded set $\{1, \ldots, 5\}$) versus $\rho$. Interestingly, we observe a phase transition in the sense that for some $\rho_*$, if $\rho < \rho_*$ then $k^* = 1$, due to the instability effect explained above, while for $\rho > \rho_*$ the regularization mitigates the stability issue and the first effect (DP noise) outweighs, making $k^* = 5$ the optimal choice. We also see that the phase transition threshold $\rho_*$ increases as $\psi$ decreases. Recall that $\psi = n/d$, and so the instability due to sample size is stronger at smaller $\psi$. Hence, a larger regularization is needed to control it. Also note that if we optimize jointly over $(k, \rho)$, the risk is minimized at $\rho = 1$ and $k = 5$, i.e., the instance-level loss with the largest possible value of $k$ in the predetermined range.

## 5 NUMERICAL EXPERIMENTS

**Numerical verification of the theory** In our first set of experiments, we corroborate our theory derived in Section 2.2 with simulations. We set $d = 100$ and generate $\boldsymbol{\theta}_0$ to have i.i.d normal entries and then scale it to be unit norm. The sample size is set as $n = \psi d$ and the noise standard deviation is set as $\sigma = \|\boldsymbol{\theta}_0\|/\text{SNR} = 1/\text{SNR}$. The solid curves in Figure 3 are the theoretical curves and the dots correspond to the simulation. Although the derived theory is in an asymptotic regime, we already see a perfect match for $d = 100$.

**Boston Housing dataset.** To investigate the optimal value of the regularization parameter $\rho$ in the interpolating loss, for different bag sizes, we conduct numerical experiments on the Boston Housing dataset. Recall that $\rho = 0$ corresponds to the bag-level loss and $\rho = 1$ corresponds to the instance-level loss. The Boston Housing dataset is a regression dataset that predicts the median value of owner-occupied homes in Boston, Massachusetts, in the mid-1970s (Harrison Jr & Rubinfeld,

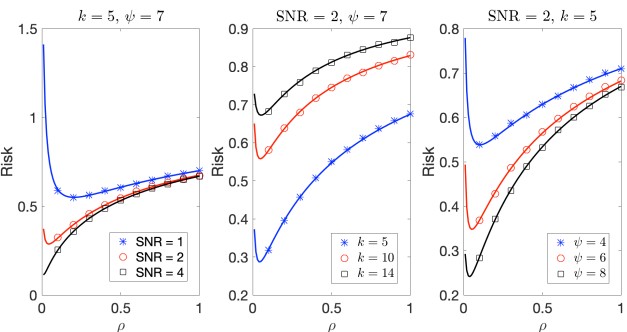

Figure 3: Solid curves correspond are the theoretical curves (Theorem 2.5) and the dots (symbols) correspond to simulations. Here $d = 100$, and we already see a perfect match between the (asymptotic) theory and the simulations.

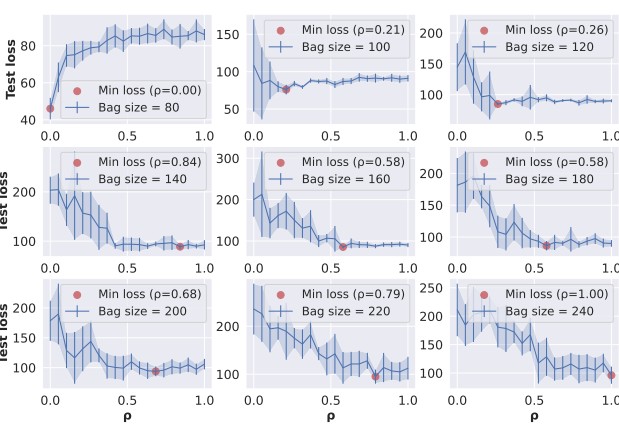

Figure 4: Test loss versus the regularization parameter $\rho$ for different bag sizes, on the Boston Housing dataset. The values on the lines represent the mean of the test loss and the error bars represent the standard deviation of the test loss. The red dots denote the optimal value of $\rho$ that achieves the minimum test loss.

1978). The dataset contains 13 features, including the crime rate, per capita income, and the number of rooms per dwelling. We use a feed-forward neural network to learn the housing prices in this dataset. The network has four hidden layers, each with 64 neurons. The activation function for all hidden layers is ReLU. The output layer has one neuron which outputs the predicted housing price.

We experiment with bag sizes ranging from 80 to 240. For each bag size, we plot the test loss against the regularization parameter $\rho$. For each value of $\rho$ in $[0, 1]$, we train 20 models and compute the mean and the standard deviation of their test loss. The results are shown in Figure 4, where the values on the lines represent the mean of the test loss and the error bars represent the standard deviation of the test loss. We also use red dots to denote the optimal value of $\rho$ that achieves the minimum test loss.

From Figure 4, we have the following observations: First, for a fixed bag size, smaller $\rho$ values (which means that the loss is closer to the bag-level loss) yield larger variance, while larger $\rho$ values (which means that the loss is closer to the instance-level loss) yield smaller variance. This confirms our theory that the instance-level loss actually has a regularization effect (as shown in Lemma 2.1). Second, as the bag size increases, the optimal $\rho$ value increases. For example, when the bag size is 80, the optimal $\rho$ value is 0, while when the bag size is 240, the optimal $\rho$ value is 1. For bag sizes between 80 and 240, the optimal $\rho$ value is achieved in $(0, 1)$. In other words, larger bag sizes require stronger regularization (larger $\rho$) to achieve the minimum test loss.

## 6 CONCLUSION

In this paper, we studied the problem of learning from aggregate responses, a privacy-preserving machine learning technique. We compared two loss functions for aggregate learning and proposed a novel interpolating estimator. We also provided a theoretical analysis of the interpolating estimator for linear regression tasks and proposed a mechanism for differentially private learning from aggregate responses. Finally, we conducted experiments to corroborate our theory and show the efficacy of the interpolating estimator.

ACKNOWLEDGEMENT

Adel Javanmard is supported in part by the NSF CAREER Award DMS-1844481 and the NSF Award DMS-2311024.

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

# A  PROOF OF THEOREMS AND TECHNICAL LEMMAS

## A.1  PROOF OF LEMMA 2.1

Recall the shorthand $\bar{y}_a = (\sum_{i \in B_a} y_i)/k$, for $a \in [m]$. We have,

$$
\begin{aligned}
\mathcal{L}_{\text{ins}}(\boldsymbol{\theta}) &= \frac{1}{mk} \sum_{a=1}^{m} \sum_{i \in B_a} (\bar{y}_a - f_{\boldsymbol{\theta}}(\boldsymbol{x}_i))^2 \\
&= \frac{1}{mk} \sum_{a=1}^{m} \sum_{i \in B_a} \Big( \bar{y}_a - \frac{1}{k} \sum_{j \in B_a} f_{\boldsymbol{\theta}}(\boldsymbol{x}_j) + \frac{1}{k} \sum_{j \in B_a} f_{\boldsymbol{\theta}}(\boldsymbol{x}_j) - f_{\boldsymbol{\theta}}(\boldsymbol{x}_i) \Big)^2 \\
&= \frac{1}{mk} \sum_{a=1}^{m} \sum_{i \in B_a} \Big( \bar{y}_a - \frac{1}{k} \sum_{j \in B_a} f_{\boldsymbol{\theta}}(\boldsymbol{x}_j) \Big)^2 + \frac{1}{mk} \sum_{a=1}^{m} \sum_{i \in B_a} \Big( \frac{1}{k} \sum_{j \in B_a} f_{\boldsymbol{\theta}}(\boldsymbol{x}_j) - f_{\boldsymbol{\theta}}(\boldsymbol{x}_i) \Big)^2 \\
&\quad + \frac{2}{mk} \sum_{a=1}^{m} \sum_{i \in B_a} \Big( \bar{y}_a - \frac{1}{k} \sum_{j \in B_a} f_{\boldsymbol{\theta}}(\boldsymbol{x}_j) \Big) \Big( \frac{1}{k} \sum_{j \in B_a} f_{\boldsymbol{\theta}}(\boldsymbol{x}_j) - f_{\boldsymbol{\theta}}(\boldsymbol{x}_i) \Big)
\end{aligned}
$$

Note that the first term can be written as

$$
\frac{1}{mk} \sum_{a=1}^{m} \sum_{i \in B_a} \Big( \bar{y}_a - \frac{1}{k} \sum_{j \in B_a} f_{\boldsymbol{\theta}}(\boldsymbol{x}_j) \Big)^2 = \frac{1}{k} \sum_{a=1}^{m} \Big( \bar{y}_a - \frac{1}{k} \sum_{j \in B_a} f_{\boldsymbol{\theta}}(\boldsymbol{x}_j) \Big)^2 = \mathcal{L}_{\text{bag}}(\boldsymbol{\theta}) \, .
$$

For the second term, we have

$$
\sum_{i \in B_a} \Big( \frac{1}{k} \sum_{j \in B_a} f_{\boldsymbol{\theta}}(\boldsymbol{x}_j) - f_{\boldsymbol{\theta}}(\boldsymbol{x}_i) \Big)^2 = \frac{1}{k} \sum_{i,j \in B_a} (f_{\boldsymbol{\theta}}(\boldsymbol{x}_i) - f_{\boldsymbol{\theta}}(\boldsymbol{x}_j))^2 = \mathcal{R}(\boldsymbol{\theta}) \, ,
$$

where we used the following identity for $a_1, \ldots, a_k$ and $\bar{a} = (\sum_{i=1}^{k} a_i)/k$:

$$
\sum_{i=1}^{k} (a_i - \bar{a})^2 = \frac{1}{k} \sum_{i,j=1}^{k} (a_i - a_j)^2 \, . \tag{11}
$$

Finally, the third term works out at zero because

$$
\begin{aligned}
&\sum_{a=1}^{m} \sum_{i \in B_a} \Big( \bar{y}_a - \frac{1}{k} \sum_{j \in B_a} f_{\boldsymbol{\theta}}(\boldsymbol{x}_j) \Big) \Big( \frac{1}{k} \sum_{j \in B_a} f_{\boldsymbol{\theta}}(\boldsymbol{x}_j) - f_{\boldsymbol{\theta}}(\boldsymbol{x}_i) \Big) \\
&= \sum_{a=1}^{m} \Big( \bar{y}_a - \frac{1}{k} \sum_{j \in B_a} f_{\boldsymbol{\theta}}(\boldsymbol{x}_j) \Big) \Big[ \sum_{i \in B_a} \Big( \frac{1}{k} \sum_{j \in B_a} f_{\boldsymbol{\theta}}(\boldsymbol{x}_j) - f_{\boldsymbol{\theta}}(\boldsymbol{x}_i) \Big) \Big] = 0,
\end{aligned}
$$

since

$$
\sum_{i \in B_a} \Big( \frac{1}{k} \sum_{j \in B_a} f_{\boldsymbol{\theta}}(\boldsymbol{x}_j) - f_{\boldsymbol{\theta}}(\boldsymbol{x}_i) \Big) = \sum_{j \in B_a} f_{\boldsymbol{\theta}}(\boldsymbol{x}_j) - \sum_{i \in B_a} f_{\boldsymbol{\theta}}(\boldsymbol{x}_i) = 0 \, .
$$

Combining the three terms together we arrive at $\mathcal{L}_{\text{ins}}(\boldsymbol{\theta}) = \mathcal{L}_{\text{bag}}(\boldsymbol{\theta}) + \mathcal{R}(\boldsymbol{\theta})$.

## A.2  PROOF OF LEMMA 2.2

We use the shorthand $\bar{f}_a = (\sum_{i \in B_a} f_{\boldsymbol{\theta}}(\boldsymbol{x}_i))/k$, for $a \in [m]$. By Taylor's expansion of the loss $\ell$ on its second argument we have

$$
\ell(\bar{y}_a, f_{\boldsymbol{\theta}}(\boldsymbol{x}_i)) = \ell(\bar{y}_a, \bar{f}_a) + \frac{\partial}{\partial b} \ell(\bar{y}_a, \bar{f}_a)(f_{\boldsymbol{\theta}}(\boldsymbol{x}_i) - \bar{f}_a) + \frac{1}{2} \frac{\partial^2}{\partial b^2} \ell(\bar{y}_a, f)(f_{\boldsymbol{\theta}}(\boldsymbol{x}_i) - \bar{f}_a)^2 \, ,
$$

for some $f$ between $\bar{f}_a$ and $f_{\boldsymbol{\theta}}(\boldsymbol{x}_i)$ and $\partial/\partial_b$, $\partial^2/\partial_b^2$ indicate the first and second derivative of $\ell(a, b)$ with respect to the second input $b$.

Summing both sides of the above equation over $i \in B_a$, the second term works out at zero since $\sum_{i \in B_a} (f_{\boldsymbol{\theta}}(\boldsymbol{x}_i) - \bar{f}_a) = 0$. Using the bound on the second derivative we arrive at

$$\sum_{i \in B_a} \ell(\bar{y}_a, f_{\boldsymbol{\theta}}(\boldsymbol{x}_i)) \leq k \ell(\bar{y}_a, \bar{f}_a) + \sum_{i \in B_a} C(f_{\boldsymbol{\theta}}(\boldsymbol{x}_i) - \bar{f}_a)^2 \, .$$

Next, summing both sides of the above equation over bags $a \in [m]$, and dividing by $mk$, we get

$$\mathcal{L}_{\text{ins}}(\boldsymbol{\theta}) \leq \mathcal{L}_{\text{bag}}(\boldsymbol{\theta}) + \frac{1}{k} \sum_{a=1}^{m} \sum_{i \in B_a} C(f_{\boldsymbol{\theta}}(\boldsymbol{x}_i) - \bar{f}_a)^2 \, .$$

By invoking identity (11) in the above we arrive at (5).

We are now ready to prove the second part of the statement. If the loss $\ell(\cdot, \cdot)$ is convex in the second input, by the Jensen's inequality we have

$$\frac{1}{k} \sum_{i \in B_a} \ell(\bar{y}_a, f_{\boldsymbol{\theta}}(\boldsymbol{x}_i)) \geq \ell\left(\bar{y}_a, \frac{1}{k} \sum_{i \in B_a} f_{\boldsymbol{\theta}}(\boldsymbol{x}_i)\right)$$

Taking the average of both side over the bags $a \in [m]$, we obtain that $\mathcal{L}_{\text{ins}}(\boldsymbol{\theta}) \geq \mathcal{L}_{\text{bag}}(\boldsymbol{\theta})$, which completes the proof of lemma.

## B    PROOF OF THEOREM 2.5

Recall $m$ as the number of bags, and $n$ as the number of samples. Since the bags are non-overlapping and each of size $k$, we have $m = n/k$. Define $\boldsymbol{S} \in \mathbb{R}^{m \times n}$, as a matrix the encodes the bagging structure, with $S_{ia} = 1/\sqrt{k} \mathbf{1}_{\{j \in B_a\}}$ where $B_a$ indicates $a$-th bag, for $a \in [m]$.

We next write the bag-level loss function and the instance-level loss function in terms of $\boldsymbol{S}$ as follows:

$$\mathcal{L}_{\text{bag}}(\boldsymbol{\theta}) = \frac{1}{km} \|\boldsymbol{S}(\boldsymbol{y} - \boldsymbol{X}\boldsymbol{\theta})\|_2^2 \, ,$$

$$\mathcal{L}_{\text{ins}}(\boldsymbol{\theta}) = \frac{1}{km} \|\boldsymbol{S}^\top \boldsymbol{S} \boldsymbol{y} - \boldsymbol{X}\boldsymbol{\theta}\|_2^2 \, .$$

The interpolating loss function (7) then reads as

$$\mathcal{L}_{\text{int}}(\boldsymbol{\theta}) = \frac{1}{mk} \left( (1-\rho)\|\boldsymbol{S}\boldsymbol{X}\boldsymbol{\theta} - \boldsymbol{S}\boldsymbol{y}\|_2^2 + \rho \left\|\boldsymbol{X}\boldsymbol{\theta} - \boldsymbol{S}^\top \boldsymbol{S}\boldsymbol{y}\right\|_2^2 \right) \, .$$

This can equivalently be written as

$$\mathcal{L}_{\text{int}}(\boldsymbol{\theta}) = \frac{1}{mk} \left\| \begin{pmatrix} \sqrt{\rho}\boldsymbol{I} \\ \sqrt{1-\rho}\boldsymbol{S} \end{pmatrix} \boldsymbol{X}\boldsymbol{\theta} - \begin{pmatrix} \sqrt{\rho}\boldsymbol{S}^\top \boldsymbol{S}\boldsymbol{y} \\ \sqrt{1-\rho}\boldsymbol{S}\boldsymbol{y} \end{pmatrix} \right\|_2^2 \, .$$

The minimizer of the above loss admits a closed-from solution given by $\widehat{\boldsymbol{\theta}}_{\text{int}} = \boldsymbol{B}\boldsymbol{y}$, with

$$\boldsymbol{B} = \left( \boldsymbol{X}^\top \left( \rho \boldsymbol{I} + (1-\rho)\boldsymbol{S}^\top \boldsymbol{S} \right) \boldsymbol{X} \right)^{-1} \boldsymbol{X}^\top \boldsymbol{S}^\top \boldsymbol{S} \, .$$

We define the shorthand $\boldsymbol{E} := \rho \boldsymbol{I} + (1-\rho)\boldsymbol{S}^\top \boldsymbol{S} \in \mathbb{R}^{n \times n}$, which is non-singular for $\rho > 0$, and $\boldsymbol{M} = (\boldsymbol{X}^\top \boldsymbol{E} \boldsymbol{X})^{-1} \boldsymbol{X}^\top \in \mathbb{R}^{d \times n}$. We then have $\boldsymbol{B} = \boldsymbol{M}\boldsymbol{S}^\top \boldsymbol{S}$.

We next recall the bias-variance decomposition (6), where the bias and variance are given by

$$\begin{aligned} \text{Bias}_{\boldsymbol{X}}(\widehat{\boldsymbol{\theta}}_{\text{int}}) &= \|(\boldsymbol{B}\boldsymbol{X} - \boldsymbol{I})\boldsymbol{\theta}_0\|_2^2 \\ &= \|(\boldsymbol{M}\boldsymbol{S}^\top \boldsymbol{S}\boldsymbol{X} - \boldsymbol{M}\boldsymbol{E}\boldsymbol{X})\boldsymbol{\theta}_0\|_2^2 \\ &= \|\boldsymbol{M}(\boldsymbol{S}^\top \boldsymbol{S} - \boldsymbol{E})\boldsymbol{X}\boldsymbol{\theta}_0\|_2^2 \, , \end{aligned} \tag{12}$$

$$\text{Var}_{\boldsymbol{X}}(\widehat{\boldsymbol{\theta}}_{\text{int}}) = \sigma^2 \|\boldsymbol{M}\boldsymbol{S}^\top \boldsymbol{S}\|_F^2 \, , \tag{13}$$

with $\|\cdot\|_F$ indicating the matrix Frobenius norm.

We continue by treating the bias and the variance separately.

### B.1 CALCULATING THE BIAS

Since the distribution of the features matrix $\boldsymbol{X}$ is invariant under rotation, we can assume that $\boldsymbol{\theta}_0 = \|\boldsymbol{\theta}_0\|\boldsymbol{e}_i$, where $\boldsymbol{e}_i \in \mathbb{R}^d$ is the vector with one at $i$-th entry and zero everywhere else. By taking average on $i \in [d]$ we obtain

$$
\begin{aligned}
\mathsf{Bias}_{\boldsymbol{X}}(\widehat{\boldsymbol{\theta}}_{\mathrm{int}}) &\stackrel{(d)}{=} \frac{\|\boldsymbol{\theta}_0\|_2^2}{d} \sum_{i\in[d]} \|\boldsymbol{M}(\boldsymbol{S}^\top\boldsymbol{S} - \boldsymbol{E})\boldsymbol{X}\boldsymbol{e}_i\|_2^2 \\
&= \frac{\|\boldsymbol{\theta}_0\|_2^2}{d} \operatorname{tr}\left(\boldsymbol{M}(\boldsymbol{S}^\top\boldsymbol{S} - \boldsymbol{E})\boldsymbol{X}\Big(\sum_{i\in[p]} \boldsymbol{e}_i\boldsymbol{e}_i^\top\Big)\boldsymbol{X}^\top(\boldsymbol{S}^\top\boldsymbol{S} - \boldsymbol{E})\boldsymbol{M}^\top\right) \\
&= \frac{\|\boldsymbol{\theta}_0\|_2^2}{d}\left\|\boldsymbol{M}(\boldsymbol{S}^\top\boldsymbol{S} - \boldsymbol{E})\boldsymbol{X}\right\|_F^2.
\end{aligned}
$$

Let us define $\boldsymbol{\Lambda} \in \mathbb{R}^{n\times n}$ as follows:

$$
\begin{aligned}
\boldsymbol{\Lambda} &:= -(\boldsymbol{S}^\top\boldsymbol{S} - \boldsymbol{E}) \\
&= -(\boldsymbol{S}^\top\boldsymbol{S} - (\rho\boldsymbol{I} + (1-\rho)\boldsymbol{S}^\top\boldsymbol{S})) \\
&= \rho(\boldsymbol{I} - \boldsymbol{S}^\top\boldsymbol{S}).
\end{aligned}
\tag{14}
$$

The bias can then be written in terms of $\boldsymbol{\Lambda}$ as $\mathsf{Bias}_{\boldsymbol{X}}(\boldsymbol{\theta}) = \frac{1}{d}\|\boldsymbol{M}\boldsymbol{\Lambda}\boldsymbol{X}\|_F^2$. In our next lemma, we characterize the asymptotic behavior of the bias.

**Lemma B.1.** *Under the asymptotic regime of Assumption 2.3, we have*

$$
\frac{1}{d}\|\boldsymbol{M}\boldsymbol{\Lambda}\boldsymbol{X}\|_F^2 \stackrel{(p)}{\to} \alpha_*^2 + \frac{\alpha_*^2}{\frac{(k-1)\psi}{k^2(1-\alpha_*)^2} - \left(\frac{\alpha_*}{1-\alpha_*}\right)^2\frac{1}{k} - \frac{k-1}{k}},
$$

*where $\alpha_*$ is the nonnegative fixed point of the following equation:*

$$
\rho + \frac{\psi}{k(1-\alpha_*)} - 1 = \frac{\psi}{k\alpha_*}\rho(k-1).
$$

Since $\|\boldsymbol{\theta}_0\| \to 1$, the result (8) follows from Lemma B.1.

We refer to the supplementary D.1 for the proof of Lemma B.1.

### B.2 CALCULATING THE VARIANCE

Since the bags are non-overlapping we have $\boldsymbol{S}\boldsymbol{S}^\top = \boldsymbol{I}_m$. Therefore $\boldsymbol{S}^\top\boldsymbol{S}$ is a projection matrix and can be written as $\boldsymbol{S}^\top\boldsymbol{S} = \boldsymbol{U}\boldsymbol{U}^\top$, with $\boldsymbol{U} \in \mathbb{R}^{n\times m}$ an orthogonal matrix. Recall that the variance is given by $\mathsf{Var}_{\boldsymbol{X}}(\widehat{\boldsymbol{\theta}}_{\mathrm{int}}) = \sigma^2\|\boldsymbol{M}\boldsymbol{S}^\top\boldsymbol{S}\|_F^2$. We use the next lemma to characterize the asymptotic behavior of the variance.

**Lemma B.2.** *Under the asymptotic regime of Assumption 2.3, for any vector $\boldsymbol{a} \in \mathbb{R}^m$ we have*

$$
\frac{n}{\|\boldsymbol{a}\|^2}\|\boldsymbol{M}\boldsymbol{U}\boldsymbol{a}\|_2^2 \stackrel{(p)}{\to} \frac{k}{v_*},
$$

*where $v_*$ is given as the fixed point of the following system of equations in $(v, u)$:*

$$
\begin{cases}
\frac{\psi}{1+u} + \frac{\rho\psi(k-1)}{\rho+u} &= k, \\
\frac{\psi(1+v)}{(1+u)^2} + \frac{\rho^2\psi(k-1)}{(\rho+u)^2} &= k.
\end{cases}
$$

Proof of Lemma B.2 is given in the supplementary D.2.

We next use the above lemma for each row of $\boldsymbol{U}$ separately (as the vector $\boldsymbol{a}$) and add them together.

Using the fact that $\|\boldsymbol{U}\|_F^2 = m$ and $m/n = k$, we get that $\|\boldsymbol{M}\boldsymbol{U}\boldsymbol{U}^\top\|_F^2 \stackrel{(p)}{\to} 1/v_*$, which completes the variance calculation.

### B.3 PROOF OF LEMMA 4.2

We use the idea of (Dwork et al., 2014, Theorem 3.6) to prove this lemma. Given a database $D = (y_1, y_2, \ldots, y_n)$, Algorithm 1 (we denote this mapping by $\mathcal{A} : \mathbb{R}^n \to \mathbb{R}^m$) outputs $m$ real numbers $(\tilde{y}_1, \tilde{y}_2, \ldots, \tilde{y}_m)$. Given the database $D$, we define the map $f : \mathbb{R}^n \to \mathbb{R}^m$ by $D \mapsto (\bar{y}_1, \bar{y}_2, \ldots, \bar{y}_m)$, which computes the mean of labels in each bag. Fix any pair of neighboring databases $D, D'$ that differ in the label of a single example. We have $\|f(D) - f(D')\|_1 := \sum_{a \in [m]} |f(D)_a - f(D')_a| \le \Delta f := \frac{C\sqrt{\log n}}{k}$. In this argument, we used Assumption 2.4 that assumes non-overlapping bags, and therefore, changing a certain $y_i$ in $D$ leads to a change in only one of $\bar{y}_a$ by at most $\Delta f$. Let $p_{\mathcal{A}(D)}(z)$ and $p_{\mathcal{A}(D')}(z)$ denote the probability density function of $\mathcal{A}(D)$ and $\mathcal{A}(D')$. We have

$$
\begin{aligned}
\frac{p_{\mathcal{A}(D)}(z)}{p_{\mathcal{A}(D')}(z)} &= \prod_{a \in [m]} \frac{\exp\left(-\frac{\varepsilon|f(D)_a - z_a|}{\Delta f}\right)}{\exp\left(-\frac{\varepsilon|f(D')_a - z_a|}{\Delta f}\right)} \\
&= \prod_{a \in [m]} \exp\left(\frac{\varepsilon\left(|f(D')_a - z_a| - |f(D)_a - z_a|\right)}{\Delta f}\right) \\
&\le \prod_{a \in [m]} \exp\left(\frac{\varepsilon|f(D')_a - f(D)_a|}{\Delta f}\right) \\
&= \exp\left(\frac{\varepsilon\|f(D) - f(D')\|_1}{\Delta f}\right) \\
&\le e^\varepsilon,
\end{aligned}
$$

which completes the proof.

## C PROOF OF THEOREM 4.3

Recall that in Algorithm 1, the individual responses are first truncated by $C\sqrt{\log n}$ and then after the aggregate responses are computed, a Laplace noise is added to them to ensure label DP. We define $\mathcal{E}$ is the event that no truncation happens, namely:

$$
\mathcal{E} := \mathbf{1}_{\{|y_i| \le C\sqrt{\log n}, \, \forall i \in [n]\}}. \tag{15}
$$

Since $y_i \sim N(0, \|\boldsymbol{\theta}_0\|^2 + \sigma^2)$, $\|\boldsymbol{\theta}_0\| = 1$, by using Gaussian tail bound along with union bounding we arrive at

$$
\mathbb{P}(\mathcal{E}) \ge 1 - n \exp\left(-\frac{C^2}{2(1 + \sigma^2)}\log n\right) = 1 - n^{-c}, \tag{16}
$$

with $c = \frac{C^2}{2(1+\sigma^2)} - 1 > 0$.

We next bound the risk of estimator $\widehat{\boldsymbol{\theta}}_{\mathrm{int}}$ as follows:

$$
\mathsf{Risk}_{\boldsymbol{X}}(\widehat{\boldsymbol{\theta}}_{\mathrm{int}}) = \mathbb{E}[\|\widehat{\boldsymbol{\theta}}_{\mathrm{int}} - \boldsymbol{\theta}_0\|^2 \mathbf{1}_{\{\mathcal{E}\}}|\boldsymbol{X}] + \mathbb{E}[\|\widehat{\boldsymbol{\theta}}_{\mathrm{int}} - \boldsymbol{\theta}_0\|^2 \mathbf{1}_{\{\mathcal{E}^c\}}|\boldsymbol{X}]. \tag{17}
$$

For the first term, note that on the instance $\mathcal{E}$ (no truncation), the privatized aggregate responses are just the aggregate responses with an additive zero mean noise with variance $2C^2 \log n/(k\varepsilon)^2$. So we can use the analysis in the proof of Theorem 2.5 with the inflated noise variance. Let $\widehat{\boldsymbol{\theta}}_{\mathrm{int}}^{\mathrm{nt}}$ be the estimator using untruncated responses in Algorithm 1. We then have This gives us

$$
\begin{aligned}
\frac{1}{\log n}\mathbb{E}[\|\widehat{\boldsymbol{\theta}}_{\mathrm{int}} - \boldsymbol{\theta}_0\|^2 \mathbf{1}_{\{\mathcal{E}\}}|\boldsymbol{X}] &= \frac{1}{\log n}\mathbb{E}[\|\widehat{\boldsymbol{\theta}}_{\mathrm{int}}^{\mathrm{nt}} - \boldsymbol{\theta}_0\|^2 \mathbf{1}_{\{\mathcal{E}\}}|\boldsymbol{X}] \\
&= \frac{1}{\log n}\mathbb{E}[\|\widehat{\boldsymbol{\theta}}_{\mathrm{int}}^{\mathrm{nt}} - \boldsymbol{\theta}_0\|^2|\boldsymbol{X}] - \frac{1}{\log n}\mathbb{E}[\|\widehat{\boldsymbol{\theta}}_{\mathrm{int}}^{\mathrm{nt}} - \boldsymbol{\theta}_0\|^2 \mathbf{1}_{\{\mathcal{E}^c\}}|\boldsymbol{X}] \\
&= \frac{1}{\log n}\mathsf{Bias}_{\boldsymbol{X}}(\widehat{\boldsymbol{\theta}}_{\mathrm{int}}^{\mathrm{nt}}) + \frac{1}{\log n}\mathsf{Var}_{\boldsymbol{X}}(\widehat{\boldsymbol{\theta}}_{\mathrm{int}}^{\mathrm{nt}}) - \frac{1}{\log n}\mathbb{E}[\|\widehat{\boldsymbol{\theta}}_{\mathrm{int}}^{\mathrm{nt}} - \boldsymbol{\theta}_0\|^2 \mathbf{1}_{\{\mathcal{E}^c\}}|\boldsymbol{X}],
\end{aligned} \tag{18}
$$

where the bias is given by (8) and variance is given by (9), where $\sigma^2/k$ is replaced with the inflated variance $\sigma^2/k + 2C^2 \log n/(k\varepsilon)^2$. Since $\mathsf{Bias}_X(\widehat{\boldsymbol{\theta}}_{\text{int}}^{\text{nt}})$ has a finite limit, the first term above vanishes as $n \to \infty$. For the second term we have

$$\frac{1}{\log n}\mathsf{Var}_X(\widehat{\boldsymbol{\theta}}_{\text{int}}^{\text{nt}}) \overset{(p)}{\to} \frac{2C^2}{k\varepsilon^2}\frac{1}{v_*}\,.$$

since $\sigma^2/\log n \to 0$. For the third term, by Cauchy–Schwarz inequality we have

$$\mathbb{E}[\|\widehat{\boldsymbol{\theta}}_{\text{int}}^{\text{nt}} - \boldsymbol{\theta}_0\|^2 \mathbf{1}_{\{\mathcal{E}^c\}}|X] \le \mathbb{E}[\|\widehat{\boldsymbol{\theta}}_{\text{int}}^{\text{nt}} - \boldsymbol{\theta}_0\|^4|X]^{1/2}\,\mathbb{P}(\mathcal{E}^c)\,. \tag{19}$$

Using the high probability bound on the minimum singular value of the Gaussian matrix $X$ (Vershynin, 2018, Theorem 4.6.1), we can show that $\mathbb{E}[\|\widehat{\boldsymbol{\theta}}_{\text{int}}^{\text{nt}} - \boldsymbol{\theta}_0\|^4|X]$ is bounded in probability and since $\mathbb{P}(\mathcal{E}^c) \le n^{-c}$, we conclude that the third term in (18) also vanishes as $n \to \infty$, in probability. Combining these together we arrive at

$$\frac{1}{\log n}\mathbb{E}[\|\widehat{\boldsymbol{\theta}}_{\text{int}} - \boldsymbol{\theta}_0\|^2 \mathbf{1}_{\{\mathcal{E}\}}|X] \overset{(p)}{\to} \frac{2C^2}{k\varepsilon^2}\frac{1}{v_*}\,. \tag{20}$$

Similar to (19) we can also show that

$$\frac{1}{\log n}\mathbb{E}[\|\widehat{\boldsymbol{\theta}}_{\text{int}} - \boldsymbol{\theta}_0\|^2 \mathbf{1}_{\{\mathcal{E}^c\}}|X] \overset{(p)}{\to} 0\,,$$

which along with (20) and (17) implies that

$$\frac{1}{\log n}\mathsf{Risk}_X(\widehat{\boldsymbol{\theta}}_{\text{int}}) \overset{(p)}{\to} \frac{2C^2}{k\varepsilon^2}\frac{1}{v_*}\,,$$

completing the proof.

## D  PROOF OF INTERMEDIATE LEMMAS

### D.1  PROOF OF LEMMA B.1

Write $X = [\boldsymbol{x}_1, \dots, \boldsymbol{x}_d]$ with $\boldsymbol{x}_i$ representing the $i$-th column. We then have $\|M\Lambda X\|^2 = \sum_{i=1}^d \|M\Lambda \boldsymbol{x}_i\|^2$. We compute the asymptotic behavior of each of the summand separately. Indeed, by symmetry of the distributions of $\boldsymbol{x}_i$, we will see that all summands converge to the same limit.

Recall that $M = (X^\top E X)^{-1}X^\top$. Consider the following optimization problem:

$$\boldsymbol{\alpha}_i = \arg\min_{\boldsymbol{\alpha}\in\mathbb{R}^d} \frac{1}{d}\|E^{1/2}X\boldsymbol{\alpha} - E^{-1/2}\Lambda\boldsymbol{x}_i\|_2^2\,. \tag{21}$$

It is easy to see that by the KKT condition, $\boldsymbol{\alpha}_i = (X^\top E X)^{-1}X^\top \Lambda\boldsymbol{x}_i = M\Lambda\boldsymbol{x}_i$. Therefore, we are interested in characterizing $\|\boldsymbol{\alpha}_i\|$ in the asymptotic regime, described in Assumption 2.3.

We write $\boldsymbol{\alpha}$ as $(\alpha_i, \boldsymbol{\alpha}_{\sim i})$ to separate its $i$-th entry form the rest. Likewise we write $X = [\boldsymbol{x}_i\, X_{\sim i}]$ to separate the $i$-th columns from the rest. We then have

$$\min_{\boldsymbol{\alpha}\in\mathbb{R}^d} \frac{1}{d}\|E^{1/2}X\boldsymbol{\alpha} - E^{-1/2}\Lambda\boldsymbol{x}_i\|_2^2$$

$$= \min_{\boldsymbol{\alpha}\in\mathbb{R}^p} \frac{1}{d}\|E^{1/2}\boldsymbol{x}_i\alpha_i + E^{1/2}X_{\sim i}\boldsymbol{\alpha}_{\sim i} - E^{-1/2}\Lambda\boldsymbol{x}_i\|_2^2$$

$$= \min_{\boldsymbol{\alpha}\in\mathbb{R}^d} \frac{1}{d}\|E^{1/2}X_{\sim i}\boldsymbol{\alpha}_{\sim i} + (\alpha_i E^{1/2} - E^{-1/2}\Lambda)\boldsymbol{x}_i\|_2^2$$

$$= \min_{\boldsymbol{\alpha}\in\mathbb{R}^d} \max_{\boldsymbol{v}\in\mathbb{R}^n} \frac{2}{d}\left(\boldsymbol{v}^\top(\alpha_i E^{1/2} - E^{-1/2}\Lambda)\boldsymbol{x}_i + \boldsymbol{v}^\top E^{1/2}X_{\sim i}\boldsymbol{\alpha}_{\sim i} - \frac{1}{2}\|\boldsymbol{v}\|_2^2\right)\,, \tag{22}$$

where in the last step we used the identity $\max_{\boldsymbol{v}}(\boldsymbol{v}^\top\boldsymbol{x} - \|\boldsymbol{v}\|^2/2) = \|\boldsymbol{x}\|^2/2$ for any vector $\boldsymbol{x}$.

We next note that $\boldsymbol{S}\boldsymbol{S}^\top = \boldsymbol{I}$ since the bags are non-overlapping. Therefore we can write $\boldsymbol{S}^\top \boldsymbol{S} = \boldsymbol{U}\boldsymbol{U}^\top$ for an orthogonal matrix $\boldsymbol{U} \in \mathbb{R}^{n \times m}$. We then have

$$\boldsymbol{E} := \rho\boldsymbol{I} + (1-\rho)\boldsymbol{S}^\top \boldsymbol{S} = \boldsymbol{U}\boldsymbol{U}^\top + \rho\boldsymbol{U}_\perp \boldsymbol{U}_\perp^\top, \quad \boldsymbol{\Lambda} = \rho(\boldsymbol{I} - \boldsymbol{S}^\top \boldsymbol{S}) = \rho\boldsymbol{U}_\perp \boldsymbol{U}_\perp^\top.$$

where $\boldsymbol{U}_\perp$ is an orthogonal matrix representing the orthogonal space to the column space of $\boldsymbol{U}$. We next decompose the vector $\boldsymbol{v}$ in the above optimization as $\boldsymbol{v} = \boldsymbol{U}\boldsymbol{v}_1 + \boldsymbol{U}_\perp \boldsymbol{v}_2$ and therefore $\|\boldsymbol{v}\|^2 = \|\boldsymbol{v}_1\|^2 + \|\boldsymbol{v}_2\|^2$.

We introduce the change of variable $\tilde{\boldsymbol{v}} = \boldsymbol{E}^{1/2}\boldsymbol{v}$ in optimization (22). Note that $\tilde{\boldsymbol{v}} = \boldsymbol{U}\boldsymbol{v}_1 + \sqrt{\rho}\boldsymbol{U}_\perp \boldsymbol{v}_2$. Continuing with (22) in terms of $\tilde{\boldsymbol{v}}$ we have

$$\min_{\boldsymbol{\alpha} \in \mathbb{R}^d} \max_{\tilde{\boldsymbol{v}} \in \mathbb{R}^n} \frac{2}{d} \left( \tilde{\boldsymbol{v}}^\top (\alpha_i \boldsymbol{I} - \boldsymbol{E}^{-1}\boldsymbol{\Lambda})\boldsymbol{x}_i + \tilde{\boldsymbol{v}}^\top \boldsymbol{X}_{\sim i}\boldsymbol{\alpha}_{\sim i} - \frac{1}{2}\|\boldsymbol{E}^{-1/2}\tilde{\boldsymbol{v}}\|_2^2 \right). \tag{23}$$

To analyze the asymptotic behavior of the solution to the above minimax optimization, we use the Convex-Gaussian-Minimax-Theorem (CGMT) (Thrampoulidis et al., 2015, Theorem 3), which is a power extension of the classical Gordon's Gaussian min-max theorem Gordon (1988), under additional convexity assumptions. According to CGMT, the above optimization is equivalent to the following auxiliary optimization problem:

$$\min_{\boldsymbol{\alpha} \in \mathbb{R}^d} \max_{\tilde{\boldsymbol{v}} \in \mathbb{R}^n} \frac{2}{d} \left( \tilde{\boldsymbol{v}}^\top (\alpha_i \boldsymbol{I} - \boldsymbol{E}^{-1}\boldsymbol{\Lambda})\boldsymbol{x}_i + \|\boldsymbol{\alpha}_{\sim i}\|\tilde{\boldsymbol{v}}^\top \boldsymbol{g} + \|\tilde{\boldsymbol{v}}\|\boldsymbol{h}^\top \boldsymbol{\alpha}_{\sim i} - \frac{1}{2}\|\boldsymbol{E}^{-1/2}\tilde{\boldsymbol{v}}\|_2^2 \right), \tag{24}$$

with $\boldsymbol{g} \sim N(0, \boldsymbol{I}_n)$ and $\boldsymbol{h} \sim N(0, \boldsymbol{I}_{d-1})$ independent Gaussian vectors. We next write the above optimization in terms of the components $\boldsymbol{v}_1$ and $\boldsymbol{v}_2$, noting that $\boldsymbol{E}^{-1}\boldsymbol{\Lambda} = \boldsymbol{U}_\perp \boldsymbol{U}_\perp^\top$, as follows:

$$\min_{\boldsymbol{\alpha} \in \mathbb{R}^d} \max_{\boldsymbol{v}_1, \boldsymbol{v}_2 \in \mathbb{R}^n} \frac{2}{d} \Big( \alpha_i \boldsymbol{v}_1^\top \boldsymbol{U}^\top \boldsymbol{x}_i + \sqrt{\rho}\boldsymbol{v}_2^\top \boldsymbol{U}_\perp^\top (\alpha_i \boldsymbol{I} - \boldsymbol{U}_\perp \boldsymbol{U}_\perp^\top)\boldsymbol{x}_i + \|\boldsymbol{\alpha}_{\sim i}\|(\boldsymbol{v}_1^\top \boldsymbol{U}^\top \boldsymbol{g} + \sqrt{\rho}\boldsymbol{v}_2^\top \boldsymbol{U}_\perp^\top \boldsymbol{g})$$
$$+ \sqrt{\|\boldsymbol{v}_1\|^2 + \rho\|\boldsymbol{v}_2\|^2}\boldsymbol{h}^\top \boldsymbol{\alpha}_{\sim i} - \frac{1}{2}\|\boldsymbol{v}_1\|^2 - \frac{1}{2}\|\boldsymbol{v}_2\|^2 \Big). \tag{25}$$

Define the shorthand

$$\boldsymbol{x}_1 := \boldsymbol{U}^\top \boldsymbol{x}_i \sim N(0, \boldsymbol{I}_m),$$
$$\boldsymbol{x}_2 := \boldsymbol{U}_\perp^\top \boldsymbol{x}_i \sim N(0, \boldsymbol{I}_{n-m}),$$
$$\boldsymbol{g}_1 := \boldsymbol{U}^\top \boldsymbol{g} \sim N(0, \boldsymbol{I}_m),$$
$$\boldsymbol{g}_2 := \boldsymbol{U}_\perp^\top \boldsymbol{g} \sim N(0, \boldsymbol{I}_{n-m}).$$

Then optimization (25) can be rewritten as

$$\min_{\boldsymbol{\alpha} \in \mathbb{R}^d} \max_{\boldsymbol{v}_1, \boldsymbol{v}_2 \in \mathbb{R}^n} \frac{2}{d} \Big( \alpha_i \boldsymbol{v}_1^\top \boldsymbol{x}_1 + \sqrt{\rho}(\alpha_i - 1)\boldsymbol{v}_2^\top \boldsymbol{x}_2 + \|\boldsymbol{\alpha}_{\sim i}\|(\boldsymbol{v}_1^\top \boldsymbol{g}_1 + \sqrt{\rho}\boldsymbol{v}_2^\top \boldsymbol{g}_2)$$
$$+ \sqrt{\|\boldsymbol{v}_1\|^2 + \rho\|\boldsymbol{v}_2\|^2}\boldsymbol{h}^\top \boldsymbol{\alpha}_{\sim i} - \frac{1}{2}\|\boldsymbol{v}_1\|^2 - \frac{1}{2}\|\boldsymbol{v}_2\|^2 \Big). \tag{26}$$

We fix $\|\boldsymbol{v}_1\| = \beta_1$ and $\|\boldsymbol{v}_2\| = \beta_2$ and first optimize over the directions of $\boldsymbol{v}_1$, $\boldsymbol{v}_2$ and then over the norms $\beta_1$ and $\beta_2$. This brings us to

$$\min_{\boldsymbol{\alpha} \in \mathbb{R}^d} \max_{\beta_1, \beta_2 \geq 0} \frac{2}{d} \Big( \beta_1 \|\alpha_i \boldsymbol{x}_1 + \|\boldsymbol{\alpha}_{\sim i}\|\boldsymbol{g}_1\| + \beta_2 \|\sqrt{\rho}(\alpha_i - 1)\boldsymbol{x}_2 + \|\boldsymbol{\alpha}_{\sim i}\|\sqrt{\rho}\boldsymbol{g}_2\|$$
$$+ \sqrt{\beta_1^2 + \rho\beta_2^2}\boldsymbol{h}^\top \boldsymbol{\alpha}_{\sim i} - \frac{1}{2}\beta_1^2 - \frac{1}{2}\beta_2^2 \Big). \tag{27}$$

In order to optimize over $\boldsymbol{\alpha}_{\sim i}$, we first fix its norm to $\eta := \|\boldsymbol{\alpha}_{\sim i}\|$ and optimize over its direction, and then optimize over $\eta$, which results in:

$$\min_{\eta \geq 0, \alpha_i} \max_{\beta_1, \beta_2 \geq 0} \frac{2}{d} \Big( \beta_1 \|\alpha_i \boldsymbol{x}_1 + \eta\boldsymbol{g}_1\| + \beta_2 \|\sqrt{\rho}(\alpha_i - 1)\boldsymbol{x}_2 + \eta\sqrt{\rho}\boldsymbol{g}_2\|$$
$$+ \eta\sqrt{\beta_1^2 + \rho\beta_2^2}\|\boldsymbol{h}\| - \frac{1}{2}\beta_1^2 - \frac{1}{2}\beta_2^2 \Big). \tag{28}$$

The next step in the CGMT framework is to compute the pointwise limit of the objective functions. Using the concentration of Lipschitz functions of Gaussian vectors we have

$$\frac{1}{\sqrt{d}}\|\alpha_i \boldsymbol{x}_1 + \eta \boldsymbol{g}_1\| \overset{(p)}{\to} \sqrt{(\alpha_i^2 + \eta^2)\frac{\psi}{k}},$$

$$\frac{1}{\sqrt{d}}\|\sqrt{\rho}(\alpha_i - 1)\boldsymbol{x}_2 + \eta\sqrt{\rho}\boldsymbol{g}_2\| \overset{(p)}{\to} \sqrt{(\rho(\alpha_i - 1)^2 + \rho\eta^2)\psi\left(1 - \frac{1}{k}\right)},$$

where we used Assumption 2.3, by which $n/d \to \psi$ and $m = n/k$.

We also have $\frac{1}{\sqrt{d}}\|\boldsymbol{h}\| \overset{(p)}{\to} 1$.

We therefore arrive at the following deterministic optimization problem

$$\min_{\eta \geq 0, \alpha_i} \max_{\beta_1, \beta_2 \geq 0} \left( \beta_1 \sqrt{(\alpha_i^2 + \eta^2)\frac{\psi}{k}} + \beta_2 \sqrt{(\rho(\alpha_i - 1)^2 + \rho\eta^2)\psi\left(1 - \frac{1}{k}\right)} \right.$$
$$\left. + \sqrt{\beta_1^2 + \rho\beta_2^2} - \frac{1}{2}\beta_1^2 - \frac{1}{2}\beta_2^2 \right), \tag{29}$$

where we made the change of variables $2\beta_1/\sqrt{d} \to \beta_1$ and $2\beta_2/\sqrt{d} \to \beta_2$.

By writing the stationary conditions for the above optimization, and simplifying the resulting system of equations by solving for $\beta_1$, $\beta_2$, and substituting for them in the other two equations, we arrive at the following two equations for $\alpha_i$ and $\eta$:

$$\begin{cases} \rho + \frac{\psi}{k(1-\alpha_*)} - 1 = \frac{\psi}{k\alpha_*}\rho(k-1) \\ \eta_*^2 + \frac{k\alpha_*^2}{k-1} + \frac{\eta_*^2\alpha_*^2}{(1-\alpha_*)^2(k-1)} = \frac{\psi}{k}\frac{\eta_*^2}{(1-\alpha_*)^2} \,. \end{cases}$$

As the final step, recall that by definition $\eta := \|\boldsymbol{\alpha}_{\sim i}\|$ and therefore, $\|\boldsymbol{\alpha}_i\|^2 \overset{(p)}{\to} \alpha_*^2 + \eta_*^2$. As we see it is independent of the index $i$ and therefore,

$$\frac{1}{d}\|\boldsymbol{M\Lambda X}\|^2 = \frac{1}{d}\sum_{i=1}^{d}\|\boldsymbol{M\Lambda x}_i\|^2 = \frac{1}{d}\sum_{i=1}^{d}\alpha_i^2 \overset{(p)}{\to} \alpha_*^2 + \eta_*^2 \,.$$

This completes the proof.

### D.2 PROOF OF LEMMA B.2

Recall that $\boldsymbol{M} = (\boldsymbol{X}^\top \boldsymbol{EX})^{-1}\boldsymbol{X}^\top$. Consider the following optimization problem:

$$\boldsymbol{\alpha} = \arg\min_{\boldsymbol{\alpha} \in \mathbb{R}^d} \frac{1}{d}\|\boldsymbol{E}^{1/2}\boldsymbol{X\alpha} - \boldsymbol{E}^{-1/2}\boldsymbol{Ua}\|_2^2 \,. \tag{30}$$

The solution to the above optimization problem has a closed-form solution given by $\boldsymbol{\alpha} = (\boldsymbol{X}^\top \boldsymbol{EX})^{-1}\boldsymbol{X}^\top \boldsymbol{Ua} = \boldsymbol{MUa}$. So we are interested in characterizing the norm of the optimal solution to the above optimization problem.

Similar to the proof of Lemma B.1, we use the framework of CGMT to characterize $\|\boldsymbol{\alpha}\|$ in the asymptotic regime described in Assumption 2.3.

Using the identity $\|\boldsymbol{x}\|/2 = \max_{\boldsymbol{v}}(\boldsymbol{v}^\top \boldsymbol{x} - \|\boldsymbol{v}\|^2/2)$, we rewrite the above optimization as:

$$\min_{\boldsymbol{\alpha} \in \mathbb{R}^d} \frac{1}{d}\|\boldsymbol{E}^{1/2}\boldsymbol{X\alpha} - \boldsymbol{E}^{-1/2}\boldsymbol{Ua}\|_2^2$$
$$= \min_{\boldsymbol{\alpha} \in \mathbb{R}^d} \max_{\boldsymbol{v} \in \mathbb{R}^n} \frac{2}{d}\left( \boldsymbol{v}^\top \boldsymbol{E}^{1/2}\boldsymbol{X\alpha} - \boldsymbol{v}^\top \boldsymbol{E}^{-1/2}\boldsymbol{Ua} - \frac{1}{2}\|\boldsymbol{v}\|_2^2 \right), \tag{31}$$

By using Convex-Gaussian-Minimax-Theorem (Thrampoulidis et al., 2015, Theorem 3), the above optimization is equivalent to the following auxiliary optimization problem:

$$\min_{\boldsymbol{\alpha} \in \mathbb{R}^d} \max_{\boldsymbol{v} \in \mathbb{R}^n} \frac{2}{d}\left( \|\boldsymbol{\alpha}\|\boldsymbol{v}^\top \boldsymbol{E}^{1/2}\boldsymbol{g} + \|\boldsymbol{E}^{1/2}\boldsymbol{v}\|\boldsymbol{h}^\top \boldsymbol{\alpha} - \boldsymbol{v}^\top \boldsymbol{E}^{-1/2}\boldsymbol{Ua} - \frac{1}{2}\|\boldsymbol{v}\|_2^2 \right), \tag{32}$$

with $\boldsymbol{g} \sim N(0, \boldsymbol{I}_n)$ and $\boldsymbol{h} \sim N(0, \boldsymbol{I}_d)$ independent Gaussian vectors.

We also recall that $\boldsymbol{S}^\top \boldsymbol{S} = \boldsymbol{U}\boldsymbol{U}^\top$ and so

$$\boldsymbol{E} := \rho \boldsymbol{I} + (1-\rho)\boldsymbol{S}^\top \boldsymbol{S} = \boldsymbol{U}\boldsymbol{U}^\top + \rho \boldsymbol{U}_\perp \boldsymbol{U}_\perp^\top ,$$

with $\boldsymbol{U}_\perp \in \mathbb{R}^{n \times (n-m)}$ denotes the orthogonal matrix, whose column space is orthogonal to the column space of $\boldsymbol{U}$. We decompose $\boldsymbol{v}$ to its component in the column space of $\boldsymbol{U}$ and $\boldsymbol{U}_\perp$ as

$$\boldsymbol{v} = \boldsymbol{U}\boldsymbol{v}_1 + \boldsymbol{U}_\perp \boldsymbol{v}_2, \quad \|\boldsymbol{v}\|^2 = \|\boldsymbol{v}_1\|^2 + \|\boldsymbol{v}_2\|^2 .$$

Therefore, $\boldsymbol{E}^{1/2}\boldsymbol{v} = \boldsymbol{U}\boldsymbol{v}_1 + \sqrt{\rho}\boldsymbol{U}_\perp \boldsymbol{v}_2$ and so the above optimization (32) can be written as

$$\min_{\boldsymbol{\alpha} \in \mathbb{R}^d} \max_{\boldsymbol{v}_1, \boldsymbol{v}_2 \in \mathbb{R}^n} \frac{2}{d}\Big( \|\boldsymbol{\alpha}\|\boldsymbol{v}_1^\top \boldsymbol{U}^\top \boldsymbol{g} + \sqrt{\rho}\|\boldsymbol{\alpha}\|\boldsymbol{v}_2^\top \boldsymbol{U}_\perp^\top \boldsymbol{g} + \sqrt{\|\boldsymbol{v}_1\|^2 + \rho\|\boldsymbol{v}_2\|^2}\boldsymbol{h}^\top \boldsymbol{\alpha}$$
$$- \boldsymbol{v}_1^\top \boldsymbol{a} - \frac{1}{2}\|\boldsymbol{v}_1\|^2 - \frac{1}{2}\|\boldsymbol{v}_2\|^2 \Big). \tag{33}$$

We next introduce the following change of variables:

$$\boldsymbol{g}_1 := \boldsymbol{U}^\top \boldsymbol{g} \sim N(0, \boldsymbol{I}_m),$$
$$\boldsymbol{g}_2 := \boldsymbol{U}_\perp^\top \boldsymbol{g} \sim N(0, \boldsymbol{I}_{n-m}).$$

Rewriting the optimization in terms of $\boldsymbol{g}_1$ and $\boldsymbol{g}_2$ we get

$$\min_{\boldsymbol{\alpha} \in \mathbb{R}^d} \max_{\boldsymbol{v}_1, \boldsymbol{v}_2 \in \mathbb{R}^n} \frac{2}{d}\Big( \|\boldsymbol{\alpha}\|\boldsymbol{v}_1^\top \boldsymbol{g}_1 + \sqrt{\rho}\|\boldsymbol{\alpha}\|\boldsymbol{v}_2^\top \boldsymbol{g}_2 + \sqrt{\|\boldsymbol{v}_1\|^2 + \rho\|\boldsymbol{v}_2\|^2}\boldsymbol{h}^\top \boldsymbol{\alpha}$$
$$- \boldsymbol{v}_1^\top \boldsymbol{a} - \frac{1}{2}\|\boldsymbol{v}_1\|^2 - \frac{1}{2}\|\boldsymbol{v}_2\|^2 \Big). \tag{34}$$

We next do the maximization on $\boldsymbol{v}_1$ and $\boldsymbol{v}_2$ by first fixing the norms to $\beta_1 := \|\boldsymbol{v}_1\|$ and $\beta_2 := \|\boldsymbol{v}_2\|$ and maximize over the directions and then maximize over $\beta_1, \beta_2$. This gives us

$$\min_{\boldsymbol{\alpha} \in \mathbb{R}^d} \max_{\beta_1, \beta_2 \geq 0} \frac{2}{d}\Big( \beta_1 \|\|\boldsymbol{\alpha}\|\boldsymbol{g}_1 - \boldsymbol{a}\| + \beta_2 \sqrt{\rho}\|\boldsymbol{\alpha}\|\|\boldsymbol{g}_2\| + \sqrt{\beta_1^2 + \rho\beta_2^2}\boldsymbol{h}^\top \boldsymbol{\alpha} - \frac{\beta_1^2 + \beta_2^2}{2} \Big). \tag{35}$$

For minimization over $\boldsymbol{\alpha}$, we first fix its norm to $\eta := \|\boldsymbol{\alpha}\|$ and optimize over its direction, and then over $\eta$:

$$\min_{\eta \geq 0} \max_{\beta_1, \beta_2 \geq 0} \frac{2}{d}\Big( \beta_1 \|\eta\boldsymbol{g}_1 - \boldsymbol{a}\| + \beta_2 \sqrt{\rho}\eta\|\boldsymbol{g}_2\| - \eta\sqrt{\beta_1^2 + \rho\beta_2^2}\,\|\boldsymbol{h}\| - \frac{\beta_1^2 + \beta_2^2}{2} \Big). \tag{36}$$

The next step in the CGMT framework is to compute the pointwise limit of the objective function. By concentration of Lipschitz functions of Gaussian vectors we have

$$\frac{1}{\sqrt{d}}\|\eta\boldsymbol{g}_1 - \boldsymbol{a}\| \overset{(p)}{\to} \sqrt{\frac{\|\boldsymbol{a}\|^2}{d} + \eta^2 \frac{\psi}{k}},$$
$$\frac{1}{\sqrt{d}}\|\boldsymbol{g}_2\| \overset{(p)}{\to} \sqrt{\psi\Big(1 - \frac{1}{k}\Big)},$$
$$\frac{1}{\sqrt{d}}\|\boldsymbol{h}\| \overset{(p)}{\to} 1,$$

where we used Assumption 2.3 by which $n/d \to \psi$, and Assumption 2.4 by which $m = n/k$. Using these limits in (36), we arrive at the following deterministic optimization problem:

$$\min_{\eta \geq 0} \max_{\beta_1, \beta_2 \geq 0} \beta_1 \sqrt{\frac{\|\boldsymbol{a}\|^2}{d} + \eta^2 \frac{\psi}{k}} + \beta_2 \sqrt{\rho}\eta\sqrt{\psi\Big(1 - \frac{1}{k}\Big)} - \eta\sqrt{\beta_1^2 + \rho\beta_2^2} - \frac{\beta_1^2 + \beta_2^2}{2} , \tag{37}$$

where we applied the change of variables $2\beta_1/\sqrt{d} \to \beta_1$ and $2\beta_2/\sqrt{d} \to \beta_2$.

In order to find the optimal solution we solve the stationary conditions. By setting derivative with respect to $\eta$ to zero we obtain

$$\frac{\beta_1 \eta \frac{\psi}{k}}{\sqrt{\frac{\|\boldsymbol{a}\|^2}{d} + \eta^2 \frac{\psi}{k}}} + \eta\sqrt{\rho\psi\Big(1 - \frac{1}{k}\Big)} - \sqrt{\beta_1^2 + \rho\beta_2^2} = 0 . \tag{38}$$

In addition by setting the derivative with respect to $\beta_1$ and $\beta_2$ to zero, we obtain

$$
\sqrt{\frac{\|\boldsymbol{a}\|^2}{d} + \eta^2 \frac{\psi}{k}} = \Big(\frac{\eta}{\sqrt{\beta_1^2 + \rho\beta_2^2}} + 1\Big)\beta_1 \,,
$$
$$
\eta\sqrt{\rho\psi\Big(1 - \frac{1}{k}\Big)} = \Big(\frac{\rho\eta}{\sqrt{\beta_1^2 + \rho\beta_2^2}} + 1\Big)\beta_2 \,. \tag{39}
$$

By substituting for $\beta_1$ and $\beta_2$ from (39) into (38) we get

$$
\frac{\eta\frac{\psi}{k}}{\eta + c} - 1 + \frac{\rho\eta\psi(1 - \frac{1}{k})}{\rho\eta + c} = 0 \,, \tag{40}
$$

where $c = \sqrt{\beta_1^2 + \rho\beta_2^2}$.

Also by substituting for $\beta_1$ and $\beta_2$ from (39) into the definition $c = \sqrt{\beta_1^2 + \rho\beta_2^2}$, we have

$$
\frac{\frac{\|\boldsymbol{a}\|^2}{d} + \eta^2\frac{\psi}{k}}{(\eta + c)^2} + \frac{\rho^2\eta^2\psi(1 - \frac{1}{k})}{(\rho\eta + c)^2} = 1 \,. \tag{41}
$$

We next make the change of variable: $c = \eta u$, and rewriting equations (40 and 41) as follows:

$$
\begin{cases}
\frac{\psi}{1+u} + \frac{\rho\psi(k-1)}{\rho+u} & = k \,, \\
\frac{\frac{k\|\boldsymbol{a}\|^2}{d\eta^2} + \psi}{(1+u)^2} + \frac{\rho^2\psi(k-1)}{(\rho+u)^2} & = k \,.
\end{cases}
$$

Defining $v := \frac{k\|\boldsymbol{a}\|^2}{\psi d\eta^2}$ we get the system of equations given in the lemma statement.

As the final step, recall that as we discussed at the beginning of the proof, $\boldsymbol{\alpha}_* = \boldsymbol{M}\boldsymbol{U}\boldsymbol{a}$. Therefore,

$$
\frac{n}{\|\boldsymbol{a}\|^2}\|\boldsymbol{M}\boldsymbol{U}\boldsymbol{a}\|_2^2 = \frac{n}{\|\boldsymbol{a}\|^2}\|\boldsymbol{\alpha}_*\|_2^2 \overset{(p)}{\to} \frac{n}{\|\boldsymbol{a}\|^2}\eta_*^2 = \frac{k}{v_*} \,,
$$

which completes the proof.

