} \text{Bias}(\widehat{\boldsymbol{\theta}}_{\text{int}}^{\text{nt}}) + \frac{1}{\log n} \text{Var}(\widehat{\boldsymbol{\theta}}_{\text{int}}^{\text{nt}}) - \frac{1}{\log n} \mathbb{E}[\|\widehat{\boldsymbol{\theta}}_{\text{int}}^{\text{nt}} - \boldsymbol{\theta}_0\|^2 \mathbf{1}_{\{\mathcal{E}^c\}} | \boldsymbol{X}],
\end{aligned}
\tag{18}
$$

where the bias is given by (8) and variance is given by (9), where $\sigma^2/k$ is replaced with the inflated variance $\sigma^2/k + 2C^2 \log n/(k\varepsilon)^2$. Since $\mathrm{Bias}(\widehat{\boldsymbol{\theta}}_{\mathrm{int}}^{\mathrm{nt}})$ has a finite limit, the first term above vanishes as $n \to \infty$. For the second term we have

$$\frac{1}{\log n}\mathrm{Var}(\widehat{\boldsymbol{\theta}}_{\mathrm{int}}^{\mathrm{nt}}) \overset{(p)}{\to} \frac{2C^2}{k\varepsilon^2}\frac{1}{v_*}.$$

since $\sigma^2/\log n \to 0$. For the third term, by Cauchy–Schwarz inequality we have

$$\mathbb{E}[\|\widehat{\boldsymbol{\theta}}_{\mathrm{int}}^{\mathrm{nt}} - \boldsymbol{\theta}_0\|^2 \mathbf{1}_{\{\mathcal{E}^c\}}|\boldsymbol{X}] \leq \mathbb{E}[\|\widehat{\boldsymbol{\theta}}_{\mathrm{int}}^{\mathrm{nt}} - \boldsymbol{\theta}_0\|^4|\boldsymbol{X}]^{1/2}\,\mathbb{P}(\mathcal{E}^c)\,. \tag{19}$$

Using the high probability bound on the minimum singular value of the Gaussian matrix $\boldsymbol{X}$ (Vershynin, 2018, Theorem 4.6.1), we can show that $\mathbb{E}[\|\widehat{\boldsymbol{\theta}}_{\mathrm{int}}^{\mathrm{nt}} - \boldsymbol{\theta}_0\|^4|\boldsymbol{X}]$ is bounded in probability and since $\mathbb{P}(\mathcal{E}^c) \leq n^{-c}$, we conclude that the third term in (18) also vanishes as $n \to \infty$, in probability. Combining these together we arrive at

$$\frac{1}{\log n}\mathbb{E}[\|\widehat{\boldsymbol{\theta}}_{\mathrm{int}} - \boldsymbol{\theta}_0\|^2 \mathbf{1}_{\{\mathcal{E}\}}|\boldsymbol{X}] \overset{(p)}{\to} \frac{2C^2}{k\varepsilon^2}\frac{1}{v_*}\,. \tag{20}$$

Similar to (19) we can also show that

$$\frac{1}{\log n}\mathbb{E}[\|\widehat{\boldsymbol{\theta}}_{\mathrm{int}} - \boldsymbol{\theta}_0\|^2 \mathbf{1}_{\{\mathcal{E}^c\}}|\boldsymbol{X}] \overset{(p)}{\to} 0\,,$$

which along with (20) and (17) implies that

$$\frac{1}{\log n}\,\mathrm{Risk}(\widehat{\boldsymbol{\theta}}_{\mathrm{int}}) \overset{(p)}{\to} \frac{2C^2}{k\varepsilon^2}\frac{1}{v_*}\,,$$

completing the proof.

# D    PROOF OF INTERMEDIATE LEMMAS

## D.1    PROOF OF LEMMA B.1

Write $\boldsymbol{X} = [\boldsymbol{x}_1, \ldots, \boldsymbol{x}_d]$ with $\boldsymbol{x}_i$ representing the $i$-th column. We then have $\|\boldsymbol{M}\boldsymbol{\Lambda}\boldsymbol{X}\|^2 = \sum_{i=1}^d \|\boldsymbol{M}\boldsymbol{\Lambda}\boldsymbol{x}_i\|^2$. We compute the asymptotic behavior of each of the summand separately. Indeed, by symmetry of the distributions of $\boldsymbol{x}_i$, we will see that all summands converge to the same limit.

Recall that $\boldsymbol{M} = (\boldsymbol{X}^\top \boldsymbol{E}\boldsymbol{X})^{-1}\boldsymbol{X}^\top$. Consider the following optimization problem:

$$\boldsymbol{\alpha}_i = \arg\min_{\boldsymbol{\alpha}\in\mathbb{R}^d} \frac{1}{d}\|\boldsymbol{E}^{1/2}\boldsymbol{X}\boldsymbol{\alpha} - \boldsymbol{E}^{-1/2}\boldsymbol{\Lambda}\boldsymbol{x}_i\|_2^2\,. \tag{21}$$

It is easy to see that by the KKT condition, $\boldsymbol{\alpha}_i = (\boldsymbol{X}^\top \boldsymbol{E}\boldsymbol{X})^{-1}\boldsymbol{X}^\top \boldsymbol{\Lambda}\boldsymbol{x}_i = \boldsymbol{M}\boldsymbol{\Lambda}\boldsymbol{x}_i$. Therefore, we are interested in characterizing $\|\boldsymbol{\alpha}_i\|$ in the asymptotic regime, described in Assumption 2.3.

We write $\boldsymbol{\alpha}$ as $(\alpha_i, \boldsymbol{\alpha}_{\sim i})$ to separate its $i$-th entry form the rest. Likewise we write $\boldsymbol{X} = [\boldsymbol{x}_i\,\boldsymbol{X}_{\sim i}]$ to separate the $i$-th columns from the rest. We then have

$$\min_{\boldsymbol{\alpha}\in\mathbb{R}^d} \frac{1}{d}\|\boldsymbol{E}^{1/2}\boldsymbol{X}\boldsymbol{\alpha} - \boldsymbol{E}^{-1/2}\boldsymbol{\Lambda}\boldsymbol{x}_i\|_2^2$$

$$= \min_{\boldsymbol{\alpha}\in\mathbb{R}^p} \frac{1}{d}\|\boldsymbol{E}^{1/2}\boldsymbol{x}_i\alpha_i + \boldsymbol{E}^{1/2}\boldsymbol{X}_{\sim i}\boldsymbol{\alpha}_{\sim i} - \boldsymbol{E}^{-1/2}\boldsymbol{\Lambda}\boldsymbol{x}_i\|_2^2$$

$$= \min_{\boldsymbol{\alpha}\in\mathbb{R}^d} \frac{1}{d}\|\boldsymbol{E}^{1/2}\boldsymbol{X}_{\sim i}\boldsymbol{\alpha}_{\sim i} + (\alpha_i\boldsymbol{E}^{1/2} - \boldsymbol{E}^{-1/2}\boldsymbol{\Lambda})\boldsymbol{x}_i\|_2^2$$

$$= \min_{\boldsymbol{\alpha}\in\mathbb{R}^d} \max_{\boldsymbol{v}\in\mathbb{R}^n} \frac{2}{d}\left(\boldsymbol{v}^\top(\alpha_i\boldsymbol{E}^{1/2} - \boldsymbol{E}^{-1/2}\boldsymbol{\Lambda})\boldsymbol{x}_i + \boldsymbol{v}^\top\boldsymbol{E}^{1/2}\boldsymbol{X}_{\sim i}\boldsymbol{\alpha}_{\sim i} - \frac{1}{2}\|\boldsymbol{v}\|_2^2\right)\,, \tag{22}$$

where in the last step we used the identity $\max_{\boldsymbol{v}}(\boldsymbol{v}^\top\boldsymbol{x} - \|\boldsymbol{v}\|^2/2) = \|\boldsymbol{x}\|^2/2$ for any vector $\boldsymbol{x}$.

We next note that $SS^\top = I$ since the bags are non-overlapping. Therefore we can write $S^\top S = UU^\top$ for an orthogonal matrix $U \in \mathbb{R}^{n \times m}$. We then have

$$E := \rho I + (1-\rho)S^\top S = UU^\top + \rho U_\perp U_\perp^\top, \quad \Lambda = \rho(I - S^\top S) = \rho U_\perp U_\perp^\top.$$

where $U_\perp$ is an orthogonal matrix representing the orthogonal space to the column space of $U$. We next decompose the vector $v$ in the above optimization as $v = Uv_1 + U_\perp v_2$ and therefore $\|v\|^2 = \|v_1\|^2 + \|v_2\|^2$.

We introduce the change of variable $\tilde{v} = E^{1/2}v$ in optimization (22). Note that $\tilde{v} = Uv_1 + \sqrt{\rho}U_\perp v_2$. Continuing with (22) in terms of $\tilde{v}$ we have

$$\min_{\alpha \in \mathbb{R}^d} \max_{\tilde{v} \in \mathbb{R}^n} \frac{2}{d}\left(\tilde{v}^\top(\alpha_i I - E^{-1}\Lambda)x_i + \tilde{v}^\top X_{\sim i}\alpha_{\sim i} - \frac{1}{2}\|E^{-1/2}\tilde{v}\|_2^2\right). \tag{23}$$

To analyze the asymptotic behavior of the solution to the above minimax optimization, we use the Convex-Gaussian-Minimax-Theorem (CGMT) (Thrampoulidis et al., 2015, Theorem 3), which is a power extension of the classical Gordon's Gaussian min-max theorem Gordon (1988), under additional convexity assumptions. According to CGMT, the above optimization is equivalent to the following auxiliary optimization problem:

$$\min_{\alpha \in \mathbb{R}^d} \max_{\tilde{v} \in \mathbb{R}^n} \frac{2}{d}\left(\tilde{v}^\top(\alpha_i I - E^{-1}\Lambda)x_i + \|\alpha_{\sim i}\|\tilde{v}^\top g + \|\tilde{v}\|h^\top \alpha_{\sim i} - \frac{1}{2}\|E^{-1/2}\tilde{v}\|_2^2\right), \tag{24}$$

with $g \sim N(0, I_n)$ and $h \sim N(0, I_{d-1})$ independent Gaussian vectors. We next write the above optimization in terms of the components $v_1$ and $v_2$, noting that $E^{-1}\Lambda = U_\perp U_\perp^\top$, as follows:

$$\min_{\alpha \in \mathbb{R}^d} \max_{v_1, v_2 \in \mathbb{R}^n} \frac{2}{d}\Big(\alpha_i v_1^\top U^\top x_i + \sqrt{\rho}v_2^\top U_\perp^\top(\alpha_i I - U_\perp U_\perp^\top)x_i + \|\alpha_{\sim i}\|(v_1^\top U^\top g + \sqrt{\rho}v_2^\top U_\perp^\top g)$$
$$+ \sqrt{\|v_1\|^2 + \rho\|v_2\|^2}h^\top \alpha_{\sim i} - \frac{1}{2}\|v_1\|^2 - \frac{1}{2}\|v_2\|^2\Big). \tag{25}$$

Define the shorthand

$$x_1 := U^\top x_i \sim N(0, I_m),$$
$$x_2 := U_\perp^\top x_i \sim N(0, I_{n-m}),$$
$$g_1 := U^\top g \sim N(0, I_m),$$
$$g_2 := U_\perp^\top g \sim N(0, I_{n-m}).$$

Then optimization (25) can be rewritten as

$$\min_{\alpha \in \mathbb{R}^d} \max_{v_1, v_2 \in \mathbb{R}^n} \frac{2}{d}\Big(\alpha_i v_1^\top x_1 + \sqrt{\rho}(\alpha_i - 1)v_2^\top x_2 + \|\alpha_{\sim i}\|(v_1^\top g_1 + \sqrt{\rho}v_2^\top g_2)$$
$$+ \sqrt{\|v_1\|^2 + \rho\|v_2\|^2}h^\top \alpha_{\sim i} - \frac{1}{2}\|v_1\|^2 - \frac{1}{2}\|v_2\|^2\Big). \tag{26}$$

We fix $\|v_1\| = \beta_1$ and $\|v_2\| = \beta_2$ and first optimize over the directions of $v_1$, $v_2$ and then over the norms $\beta_1$ and $\beta_2$. This brings us to

$$\min_{\alpha \in \mathbb{R}^d} \max_{\beta_1, \beta_2 \geq 0} \frac{2}{d}\Big(\beta_1\|\alpha_i x_1 + \|\alpha_{\sim i}\|g_1\| + \beta_2\|\sqrt{\rho}(\alpha_i - 1)x_2 + \|\alpha_{\sim i}\|\sqrt{\rho}g_2\|$$
$$+ \sqrt{\beta_1^2 + \rho\beta_2^2}h^\top \alpha_{\sim i} - \frac{1}{2}\beta_1^2 - \frac{1}{2}\beta_2^2\Big). \tag{27}$$

In order to optimize over $\alpha_{\sim i}$, we first fix its norm to $\eta := \|\alpha_{\sim i}\|$ and optimize over its direction, and then optimize over $\eta$, which results in:

$$\min_{\eta \geq 0, \alpha_i} \max_{\beta_1, \beta_2 \geq 0} \frac{2}{d}\Big(\beta_1\|\alpha_i x_1 + \eta g_1\| + \beta_2\|\sqrt{\rho}(\alpha_i - 1)x_2 + \eta\sqrt{\rho}g_2\|$$