# OpenReview forum: "Learning from Aggregate responses: Instance Level versus Bag Level Loss Functions"
_ICLR.cc/2024/Conference — ICLR 2024 poster_

### Official Review · Reviewer_2uPt · 2023-10-30

**Soundness:** 2 fair
**Presentation:** 3 good
**Contribution:** 2 fair
**Rating:** 6
**Confidence:** 3

**Summary:**

The paper studies the relationship between instance-loss and bag-loss in the setting of aggregate label. In practice, the latter would provide stronger privacy protection in the model training process. The authors showed that instance loss can be viewed as having an extra regularization term similar to the within-cluster square loss from k-means. Based on this, a interpolative loss is proposed between these two as well as the bias-variance tradeoff on this factor.

**Strengths:**

- The paper is clearly written and provides a very intuitive view on the connection between the bag-loss and individual-loss.
- The authors derived theoretical results on the SNR which has practical implications on selecting optimal bag size.

**Weaknesses:**

- The authors did not include the proof of the theorems, which I think is essential given the paper is largely centered around Theorem 2.5. In particular, (please let me know if I'm wrong), equation (8) simplifies to $2\alpha_{\ast}-1$ when k=1 which is not always 0 based on the fixed point assumption.
- There is a little incoherence between the assumption and application. The authors focused on very practical applications with DP and aggregate learning, but it's rare in those cases for the proportional regime to hold which appears to be essential to the analysis. More commonly we see d = o(n), and I also assume k could not be fixed as d grows to satisfy k-anon.

**Questions:**

As aforementioned, I would like the authors to provide the derivation of the theoretical results in the supplementary, and address the generalizability of the results if one or few assumptions do not hold in the paper.

---

> ### Author Response · Authors · 2023-11-18
>
> Thank you for taking the time to read and review our paper!
> ## Response to weaknesses:
> >The authors did not include the proof of the theorems, which I think is essential given the paper is largely centered around Theorem 2.5.
>
> We would like to clarify that we did provide detailed proofs for all the theorems in the **supplementary material**. Specifically, the proof of Theorem 2.5 can be found in Appendix B. Due to space constraints for the main manuscript, the proofs are deferred to the supplementary. It is already available to the reviewers.
> >In particular, (please let me know if I'm wrong), equation (8) simplifies to $2\alpha_*-1$ when $k=1$ which is not always 0 based on the fixed point assumption.
>
> Note that equation (8) is under the assumption $k>1$ and $\rho>0$. (So you cannot set $k=1$ in it and get a valid result). We have been specific about this: The first bullet reads: **“Bias: For $k>1$ and $\rho> 0$ ….”**. And then we add a sentence stating that under $k=1$ or $\rho=0$, the bias is zero. In other words, these two statements are the complement of each other and you should not expect to get the second as a special case of (8).
>
> >The authors focused on very practical applications with DP and aggregate learning, but it's rare in those cases for the proportional regime to hold which appears to be essential to the analysis. More commonly we see d = o(n), and I also assume k could not be fixed as d grows to satisfy k-anon.
>
> The case of $d = o(n)$ is the classical setting in statistics, while nowadays with the ability of collecting rich granular information on individuals, we are well into the proportional regime. For the practical applications discussed in the paper, there is often abundant information about users, many of which are categorical and would add even more dimension once converted into numerical features. Indeed there is a very active and broad area, called high-dimensional statistics, developed in the past three decades to discuss scenarios where $d$ exceeds sample size $n$.  We refer to (Donoho 2000) for a discussion of the relevance of the high-dimensional regime.
>
>
> Regarding the comment on $k$, we would like to recall that the setting of the paper is motivated by applications where the responses/labels contain private information, but the features vectors are not private. That’s why the aggregation is only done on the responses. That said, k-anonymity does not have anything to do with features dimension ($d$). It simply means that for each user, its aggregate response is shared with at least $k-1$ other users (since the bags are of size $k$).
> If you have further questions or if additional clarification is needed, please let us know and we will be happy to address.
>
> *Donoho (2000), High-Dimensional Data Analysis: The Curses and Blessings of Dimensionality
>
> ## Response to Question:
> Please see our response to weaknesses. The proofs are already given in the supplementary which is available to the reviewers. In particular, the proof of Theorem 2.5 can be found in Appendix B.

---

> > ### Comment · Reviewer_2uPt · 2023-11-20
> >
> > Thanks the authors' response as I missed the supplementary material initially. I'm willing to change my rating to 6: marginally above the acceptance threshold.

---

> > > ### Author Response · Authors · 2023-11-20
> > >
> > > Thank yo very much for reading the rebuttal and raising your score.

---

### Official Review · Reviewer_D7UY · 2023-10-31

**Soundness:** 3 good
**Presentation:** 3 good
**Contribution:** 2 fair
**Rating:** 6
**Confidence:** 4

**Summary:**

This paper studied the problem of learning from average response.
The author compared two loss functions (error of average output, average as instance label) theoretically and analyzed the bias and variance for linear models.
Then, the author proposed a method for differential privacy based on the theoretical result.

**Strengths:**

- This paper is well-written and easy to follow. The mathematical notation is clear.
- The author did a good job discussing real-world examples and related works.
- The analyses are sound and reasonable.
- The private aggregate learning method is a solid contribution.

**Weaknesses:**

- Most analyses are restricted to linear models. Thus, the range of applications may be limited. Even for tabular data, gradient boosting machines may perform better than linear models. There's a gap between theory and practice: for the Boston housing dataset, the author used an MLP instead.
- The experiment is limited to one dataset. The labels generated in this way may not be a good proxy of real-world aggregate label problems.
- A minor issue: the title "learning from aggregate responses" may be imprecise since here, the author only considered the average. There are other types of aggregate information, such as max, median, and rank.

**Questions:**

- Why non-overlapping?
- "The bag-level loss does not have a unique minimizer": is this proven?
- Why should we reduce the variance? It seems that the regularization could lead to an over-smoothed solution.

---

> ### Author Response · Authors · 2023-11-18
>
> Thank you for taking the time to read our paper and for your thoughtful comments.
>
> # Response to Weaknesses:
> >Most analyses are restricted to linear models. Thus, the range of applications may be limited. Even for tabular data, gradient boosting machines may perform better than linear models. There's a gap between theory and practice: for the Boston housing dataset, the author used an MLP instead.
>
> While it's true that our theoretical analyses primarily focus on linear models, by providing a precise theory for this case we develop non-trivial insights,  which as we see in the experiment carry over to more complex nonlinear setups.
>
>
> In particular, we acknowledge that real-world data often exhibits complex patterns that may be better captured by non-linear models such as gradient boosting machines or multi-layer perceptrons (MLPs). To bridge the gap between theory and practice, we conducted experiments on the Boston housing dataset using an MLP, deviating from the linear models typically analyzed in the theoretical context.
>
>
> Interestingly, our experimental results with the MLP align with the predictions of our theoretical framework. Specifically, our theory suggests that the choice between bag-level and instance-level loss depends on the signal-to-noise ratio (SNR), and our experiment on the Boston housing dataset validates this insight. When the SNR is high (indicating smaller bags with higher SNR), bag-level loss outperforms; conversely, when the SNR is low, instance-level loss is more effective. In the intermediate phase, our results reveal a sweet spot, confirming the practical relevance of our theoretical findings.
>
> >The experiment is limited to one dataset. The labels generated in this way may not be a good proxy of real-world aggregate label problems.
>
> In the setup discussed in the paper, the aggregate labels are generated by averaging the individual labels of all examples within a bag. It's worth noting that this method of label aggregation through averaging is a well-established and widely used practice. This technique finds application in various domains, from traditional group testing methodologies to contemporary privacy-preserving frameworks, such as the Private Aggregation API of Chrome Privacy Sandbox and the SKAdNetwork API from Apple.
>
> If there's a specific concern or aspect of label generation that we may not have adequately addressed in our work, we would greatly appreciate additional details or clarification. It's possible that there might be a misunderstanding, and we are keen to ensure that we address your question comprehensively.
>
> >A minor issue: the title "learning from aggregate responses" may be imprecise since here, the author only considered the average. There are other types of aggregate information, such as max, median, and rank.
>
> While we acknowledge that there are various ways to aggregate information, such as using max, median, or rank, in the literature the aggregation often refers to the average or a weighted average. This method of label aggregation through averaging is a widely adopted practice, evident in traditional group testing methodologies, as well as in contemporary privacy-preserving frameworks like the Private Aggregation API of Chrome Privacy Sandbox and the SKAdNetwork API from Apple.
>
> # Response to Questions
> >Why non-overlapping?
>
> The significance of employing non-overlapping bags is beyond considerations for our analysis; it is also crucial for maintaining privacy protection. Allowing overlapping bags, where a data example can be included in multiple bags, introduces the risk of inferring individual labels. To illustrate this potential privacy breach, consider an extreme scenario where there are more bags than the number of data examples. In such cases, an adversary could exploit the overlap to infer individual labels by solving a linear equation. Therefore, the choice of non-overlapping bags is not only for our analytical considerations but is also a deliberate measure to safeguard against potential privacy vulnerabilities. We appreciate your attention to this aspect of our methodology.
>
> >"The bag-level loss does not have a unique minimizer": is this proven?
>
> To illustrate this point, let's consider a simple example. Suppose we have a bag consisting of two examples and the aggregate response for this bag is 0.5. In this case, we can select any one of the two examples and predict its response y, and the other as 1-y. As a result, the bag-level loss on this bag becomes zero, achieving the minimum. Therefore, it does not admit a unique minimizer.
> We hope this example clarifies the observation that the bag-level loss does not admit a unique minimizer in general.

---

> > ### Author Response · Authors · 2023-11-18
> >
> > # Response to Questions (continued)
> > >Why should we reduce the variance? It seems that the regularization could lead to an over-smoothed solution.
> >
> > The rationale behind reducing variance is rooted in the classical bias-variance decomposition, a standard concept in statistics. According to this decomposition, the total risk of a model can be expressed as the sum of two components: bias and variance. By reducing variance, one can get more stability during test time and avoid overfitting.
> >
> > Of course if the regularization is unreasonably strong then it will increase the bias and so results in higher error for the model. That’s why we aim to tune the regularization parameter $\rho$ (see equation 4).

---

### Official Review · Reviewer_wg1T · 2023-11-01

**Soundness:** 3 good
**Presentation:** 2 fair
**Contribution:** 2 fair
**Rating:** 6
**Confidence:** 3

**Summary:**

This paper studies learning from aggregated responses with loss functions from both an instance level and a bag level with motivation from keeping individual privacy. It investigated the tradeoff between risk and privacy by inserting a multiplicative parameter for the regularization term $\rho$ such that the instance-level loss and bag-level loss are well balanced. It also investigated different parameters such as signal-to-noise ratio, bag size, and overparameterization, and their effects on the convergence of the bias and variance of the model, and derived the optimal choice of parameter $\rho$ with evidence from numerical experiments.

**Strengths:**

The paper investigates the trade-off between bias and variance by taking into account various factors that might influence the model performance, which is considered thorough.

**Weaknesses:**

The presentation can be improved. For example, some high-level ideas on the tradeoff could be included earlier in the paper (in more detail) instead of introducing them in the latter parts of the paper which makes it hard to grasp the contribution at the beginning. More explanation on Theorem 2.5 should be discussed. For example, the intuition of the actual meaning of the fixed point or the solution of the equation systems; what quantities the bias and the variance are converging to.

**Questions:**

Can you explain a bit more on the fixed point and the solution of the equation systems in Theorem 2.5?

**Details Of Ethics Concerns:**

No concerns.

---

> ### Author Response · Authors · 2023-11-18
>
> Thank you for taking the time to read our paper and for your thoughtful comments.
>
> # Response to Weaknesses
> >The presentation can be improved. For example, some high-level ideas on the tradeoff could be included earlier in the paper (in more detail) instead of introducing them in the latter parts of the paper which makes it hard to grasp the contribution at the beginning.
>
> Following your comment, we plan to add this explanation early on in the paper to better describe the contributions. “Note that bag-level and instance-level are approaches for training models using aggregated responses and once the bag configurations are given to the learner, she can try either loss to train her model. The high-level intuition we establish in this paper (through rigorous quantitative statements) is that using bag-level loss results in models with lower bias but higher variance compared to models that are trained by minimizing the instance-level loss. So in cases where there is a lot of heterogeneity in the responses, instance-level would work better. In contrast, for use cases with more homogenous responses, the bag-level would be better because lowering the bias becomes more effective (the variance is low).”
>
> >More explanation on Theorem 2.5 should be discussed. For example, the intuition of the actual meaning of the fixed point or the solution of the equation systems; what quantities the bias and the variance are converging to.
>
> Note that $u$ is a dummy variable and the first equation is a quadratic equation in $u$, which has one positive solution. Once it is solved in $u$, it can be plugged into the second equation which is linear in $v$, to solve for $v$.
> As we see from the result $v$ is the variable that shows up in variance formula (equation 9). We could have just written $v$ explicitly by solving for $u$ and $v$ in closed form, but thought that writing it this way would be cleaner.
>
> # Response to Question:
> >Can you explain a bit more on the fixed point and the solution of the equation systems in Theorem 2.5?
>
> Please see our response to the weaknesses as w answer this question there. If you have any other question or any further clarification is needed we are more than happy to respond.

---

> > ### Author Response · Authors · 2023-11-22
> >
> > We have responded to your comments. Please let us know if you have any other question. We will be happy to provide more clarification if needed.

---

> ### Comment · Reviewer_wg1T · 2023-11-22
> **Official comment by Reviewer wg1T**
>
> Thank you very much for addressing the concerns. It would be nice to incorporate the discussion in the revision. I would like to raise the score to 6.

---

> > ### Author Response · Authors · 2023-11-22
> >
> > Thank you for raising your score. We will definitely add the discussion to the revised version.

---

### Official Review · Reviewer_LbHA · 2023-11-06

**Soundness:** 3 good
**Presentation:** 3 good
**Contribution:** 2 fair
**Rating:** 6
**Confidence:** 3

**Summary:**

This paper investigates the challenge of learning with restricted access to labels. Specifically, the authors explore two categories: Instance Level and Bag Level Losses. The proposed algorithm begins by partitioning data points into k non-overlapping bags, each containing m examples. Subsequently, the authors introduce two sets of loss functions: bag-level (as seen in Eq. 1) and instance-level (as seen in Eq. 2).

The primary contributions of this paper are as follows:

1- Demonstrating that for a wide range of loss functions, a regularized bag-level loss can effectively serve as a surrogate for instance-level losses.

2- In the context of well-specified linear regression, providing a precise characterization of the tradeoff between bias and variance.

3- Introducing a label-DP variant of their algorithm tailored to bag-level losses.

**Strengths:**

I think the paper is well-written and the theoretical results are presented with lots of intuitions. I think the main strength of the paper is the exact characterization of the bias and variance tradeoff. This characterization let them compare two family of losses.

**Weaknesses:**

I think the motivation behind the definition of instance-level and bag-level losses is not clear. Another weakness is that the authors do not compare their results with the prior results on label DP. In this paper the authors use bag-level loss and then consider the noisy labels. Is that the best approach?

**Questions:**

1- What are the use-cases of instance-level loss and bag-level loss? To me, instance-level seems more intuitive and I want to know about the scenarios in which it is preferable to use bag-level loss.

2- What is the significance of non-overlapping bags? Is it because of the analysis?

3- For DP variants, the authors propose first forming the bag-level loss and then applying Laplace mechanism to the aggregated labels. Is that the best approach? What is the comparison of this method with the prior work on label-DP?

4- Randomness in placing the samples in bags may help in privacy amplification.

---

> ### Author Response · Authors · 2023-11-18
>
> Thank you for taking the time to read our paper and for your thoughtful comments.
>
> # Response to Weaknesses
> >I think the motivation behind the definition of instance-level and bag-level losses is not clear.
>
> Indeed having a clear understanding of these concepts is essential for comprehending our work. Note that in our setup, the learner has access to individual features vectors $x_i$, while is only given access to the aggregate responses. The instance-level and bag-level losses are two approaches often used in this area to assess the loss of a model. In the instance-level you treat the average response in a bag as a proxy for the *ungiven* individual responses in that bag and measure the loss between your model predictions $f(x_i)$ and the average response. Therefore, for each instance $x_i$ you still have a term in the loss function. In the bag-level loss, for each bag you measure how well your *average predictions* align with the given *average responses*. So for each bag you get a term in the loss function, but not for each instance.
> We hope that this explanation has been helpful. Please let us know if you have any further questions.
>
> >Another weakness is that the authors do not compare their results with the prior results on label DP. In this paper the authors use bag-level loss and then consider the noisy labels. Is that the best approach?
>
> To clarify, we do compare our results with the label DP procedure. Note that the pure label DP procedure corresponds to the choice of $k=1$, which results in clusters of size one and essentially no aggregation. In particular, in Figure 2, we discuss that if $\rho$ is small (recall that $\rho=0$ corresponds to the bag-level loss), then $k=1$ is optimal, which means that pure label DP yields better model risk under a fixed privacy budget than aggregation + DP. On the other hand, for large values of $\rho$ (recall that $\rho=1$ corresponds to the instance-level loss), larger $k$ is better, so adding noise to the aggregated responses is better than just label DP. Hence, in summary, in our experiments, we compare with pure label DP ($k=1$) and also different choices of $\rho$ (which cover both the bag-level and instance-level loss as well as any combination of them).
>
> # Response to Questions
> >1- What are the use-cases of instance-level loss and bag-level loss? To me, instance-level seems more intuitive and I want to know about the scenarios in which it is preferable to use bag-level loss.
>
> Note that bag-level and instance-level are approaches for training models using aggregated responses and once the bag configurations are given to the learner, she can try either loss to train her model. In our response to the weaknesses (above), we provided further intuition about the definition of these losses. The high-level intuition we establish in this paper (through rigorous quantitative statements) is that using bag-level loss results in models with lower bias but higher variance compared to models that are trained by minimizing the instance-level loss. So in cases where there is a lot of heterogeneity in the responses, instance-level would work better. In contrast, for use cases with more homogenous responses, the bag-level would be better because lowering the bias becomes more effective (the variance is low). For the case of linear regression, we explicitly characterize regimes where one approach is better than the other (see Lemma 3.2). In Figure 3, we also plot the risk (derived from our precise theoretical characterization). Note that the leftmost point ($\rho=0$) corresponds to the bag-level loss and the rightmost point ($\rho =1$) corresponds to the instance-level loss. As we see depending on the values of signal-to-noise ratio (SNR), cluster size (k) and overparameterization ($\psi$), one approach can outweigh the other.
>
> >2- What is the significance of non-overlapping bags? Is it because of the analysis?
>
> The significance of employing non-overlapping bags is beyond considerations for our analysis; it is also crucial for maintaining privacy protection. Allowing overlapping bags, where a data example can be included in multiple bags, introduces the risk of inferring individual labels. To illustrate this potential privacy breach, consider an extreme scenario where there are more bags than the number of data examples. In such cases, an adversary could exploit the overlap to infer individual labels by solving a linear equation. Therefore, the choice of non-overlapping bags is not only for our analytical considerations but is also a deliberate measure to safeguard against potential privacy vulnerabilities. We appreciate your attention to this aspect of our methodology.

---

> > ### Author Response · Authors · 2023-11-18
> >
> > # Response to Questions (Continued)
> > >For DP variants, the authors propose first forming the bag-level loss and then applying Laplace mechanism to the aggregated labels. Is that the best approach? What is the comparison of this method with the prior work on label-DP?
> >
> > (Please see our response to the weaknesses above): As discussed, the case of $k=1$ (clusters of size 1) corresponds to no aggregation and just pure label DP. As we show theoretically and also in Figure 4, there is a phase transition on the tuning parameter $rho$, below which $k=1$ is optimal (i.e., do not aggregation and ensure label DP by adding noise directly to individual response), and above this threshold largest allowable bag size is the best (i.e., first aggregate and then add noise to aggregated responses to ensure label DP). As we explain in the paper, the intuition is that while larger bag size results in higher model risk, they also lower the sensitivity of aggregated responses to each individual response and hence smaller noise is needed to ensure privacy. Depending on which way this trade-off goes it may be better to aggregate+DP instead of direct label DP.
> >
> > >Randomness in placing the samples in bags may help in privacy amplification.
> >
> > What matters in the analysis is that the bags are constructed independently of data. Note that under the setup discussed in the paper, the bag configurations are shared with the learner (so she knows which samples belong to which bag and forms the bag-level or instance-level loss that requires this information). Now if the bag configuration were dependent on the responses, that would already be a leak on privacy. In addition, such dependence would have introduced more bias to the model learning process.

---

> > > ### Comment · Reviewer_LbHA · 2023-11-22
> > >
> > > Thanks. I think the paper is interesting and I would like to keep my score unchanged.

---

### Official Review · Reviewer_Z7mk · 2023-11-08

**Soundness:** 4 excellent
**Presentation:** 4 excellent
**Contribution:** 2 fair
**Rating:** 6
**Confidence:** 4

**Summary:**

This paper addresses the problem of learning a model from a dataset partitioned into non-overlapping 'bags' of equal size, given the average value of the response variable in each bag. The difference with respect to usual regression analysis is that the individual responses are not provided, and so we cannot calculate the usual loss function. This calls for the definition of a modified loss function that depends only on the aggregate response in each bag.

The present paper analyses two such modified loss functions : L_bag, which measures the distance between the aggregate responses and the aggregates of model predictions ; and L_ins, which measures the distance between the aggregate responses and the individual model predictions.

While L_bag gives rise to an unbiased estimator, the estimator corresponding to L_ins has a lower variance, at the cost of introducing a bias. The author(s) interpret the difference L_ins - L_bag as a sort of regularization term.

The paper also studies a loss function L_int, depending on a parameter rho, that interpolates linearly between L_bag and L_ins. (This corresponds to tuning the strength of the regularization.) The choice of rho involves a trade-off between the bias and variance of the resulting estimator, and one can make this choice in order to minimize the 'prediction risk' of the estimator.

The paper illustrates the above ideas in the case of linear models, with a quadratic loss function. In a rather specific asymptotic regime, with the training data drawn from standard normal distributions and then assigned uniformly at random to the bags, the paper computes several quantities of interest, such as the bias and variance of the estimator corresponding to L_int (and, by specialization, of that corresponding to L_bag and L_int). The results support the qualitative conclusions mentioned above.

The paper also contains certain numerical experiments, verifying the theoretical results in the asymptotic regime mentioned above, and exploring in the case of the Boston Housing dataset the optimal value of rho as a function of bag size.

Finally, the paper also shows that truncating and adding a Laplace noise to the original aggregate responses makes the dataset epsilon-label Differentially Private. The resulting estimator is also Differentially Private, and the paper studies the optimal bag size (in the aforementioned asymptotic regime) as a function of the optimal regularization parameter rho.

**Strengths:**

A. Clarity
     1. The content is well organized. The introduction succinctly motivates the problems and summarizes the results.
     2. The intuitions are made explicit and are transparently described. This made the review process painless.
     3. The main text is clutter free and the proofs of all the technical claims are included in the supplementary material. The submission, is as a result, largely self contained and has no technical gaps.
     4. The literature review is thorough and the authors have gone the extra mile organizing the ideas and summarizing the current state of a rapidly evolving niche in ML research.

B. Significance, Quality
     1. The subject is topical since Privacy preserving ML is a burning issue and an active area of current research.
     2. The results shed new light on the interplay between instance-level and bag-level loss. It is interesting to see that the interpolate improves generalization in a real world dataset.
     3. The qualitative conclusions derived in the asymptotic regime are clarifying and may be useful in more "hairy" real world datasets.

C. Originality
     1. Mixed loss function: The idea of interpolating between the loss functions is interesting and the idea that this could result in better generalization for aggregates-based learning appears to be novel.
     2. The authors go beyond showing empirical results and derive closed form expressions and provide compelling interpretations for them.

**Weaknesses:**

1. This is a minor comment on terminology: regularizers in the ML context, to the best of this reviewer's knowledge, only depend on model parameters and not on the data (which includes the feature vectors). Shrinkage and variance-reduction are related but not identical, and regularization in ML refers to the former. If indeed this term is used more broadly it may be helpful to include a few clarifying sentences and/or references.

2. The theoretical results are proved under the following
      a. Random Bags
      b. Equal Sized Bags
      c. Non-Overlapping Bags

The introduction cites the Google(Privacy Sandbox) and Apple(SKAd) APIs as motivations for the problem but the assumptions (a),(b),(c) do not hold in this regimes. In fact, a critical feature of both (a) Google's Private Aggregation API and (b) Apple's SKAd Network is that the aggregation keys are configurable (eg. aggregate behavior of traffic from California, visiting sports content at a certain time of day) ie the bags are specified by the consumer of the API, not by random aggregations. This enables ML algorithms to effectively use these aggregates for prediction.

This is an essential part of the problem. For example, here is a quasi-technical [report published by Google](https://github.com/google/ads-privacy/blob/master/Combining%20the%20Event%20and%20Aggregate%20Summary%20Reports%20from%20the%20Privacy%20Sandbox%20Attribution%20Reporting%20API.pdf)

If there is indeed a connection/insight in the paper that this reviewer has missed, please respond to the question below.

**Questions:**

1. Are there explicit connections/insights to be gleaned from the results in the paper and the Google/Apple approaches for aggregated-feedback?

---

> ### Author Response · Authors · 2023-11-18
>
> Thank you for taking the time to read our paper and for your thoughtful comments.
>
> # Response to Weaknesses:
> >1.This is a minor comment on terminology: regularizers in the ML context ....
>
> Yes, we agree that regularization is often used to refer to techniques that only depend on model parameters. However, there is indeed a growing body of work on data-dependent regularization. Following your comments, we will add the following clarification to the paper:
> "In addition to traditional regularization techniques that only depend on model parameters, there is also a growing body of work on data-dependent regularization. This type of regularization explicitly takes into account the training data during the regularization process, which can lead to improved generalization performance in certain settings."
> We will also add references to the following papers, which provide examples of data-dependent regularization techniques:
> - Shivaswamy, P. K., & Jebara, T. (2010). Maximum Relative Margin and Data-Dependent Regularization. Journal of Machine Learning Research, 11(2).
> - Zhao, H., Tsai, Y. H. H., Salakhutdinov, R. R., & Gordon, G. J. (2019). Learning neural networks with adaptive regularization. Advances in Neural Information Processing Systems, 32.
> - Mou, W., Zhou, Y., Gao, J., & Wang, L. (2018, July). Dropout training, data-dependent regularization, and generalization bounds. In International conference on machine learning (pp. 3645-3653). PMLR.
> We hope that these additions will clarify the distinction between traditional regularization and data-dependent regularization.
>
> >2.The theoretical results are proved under the following a. Random Bags b. Equal Sized Bags c. Non-Overlapping Bags. The introduction cites the Google(Privacy Sandbox) and Apple(SKAd) APIs as motivations for the problem but the assumptions (a),(b),(c) do not hold in this regimes ....
>
> Thank you for your insightful comment. The Google Privacy Sandbox and Apple APIs are discussed in the introduction mainly to motivate the general problem of learning from aggregated data, why it is of interest in practice and the privacy protections benefits of it.  We acknowledge that the configurability of aggregation keys introduces a level of specificity that may not align perfectly with our assumption of random aggregations. However, we contend that even in scenarios where bags are specified based on certain criteria (e.g., location, content, time of day), there remains an element of randomness in the aggregation process, especially when multiple users meet the specified conditions. In this sense, our theoretical framework, while idealized, can still serve as a valuable baseline for practical considerations.
>
> Furthermore, note that constructing bags by aggregating records with specific attributes in general outperforms random bags (see Chen et al. (2023)). Therefore, our study of random bags provides a conservative estimate of performance in practical settings, acting as a benchmark for the worst-case scenario. Additionally, it's noteworthy that random bags are commonly employed in the literature on learning from aggregate labels, further emphasizing their relevance in theoretical analyses (e.g., Busa-Fekete et al. (2023), Yu et al. (2013), Quadrianto et al. (2008)).
>
> # Response to Questions
> >Are there explicit connections/insights to be gleaned from the results in the paper and the Google/Apple approaches for aggregated-feedback?
>
> Our paper aims to shed light on the trade-offs between instance-level and bag-level losses in aggregated feedback settings. While we understand that the specifics of the Google Privacy Sandbox and Apple SKAd APIs may not align directly with our theoretical framework, our findings suggest a broader principle — the potential regularization effect of instance-level loss leading to improved generalization performance. This principle may inform the design of algorithms for APIs with configurable aggregation keys, guiding the search for an optimal balance between instance-level and bag-level losses.
>
> In conclusion, we value your input and recognize the need for future work that bridges the gap between theoretical models and real-world applications. Our ongoing research will focus on extending our theoretical framework to incorporate more realistic assumptions, specifically addressing the nuances of APIs like the Google Privacy Sandbox and Apple SKAd. We are committed to exploring how our theoretical insights can be practically applied in these settings and welcome further discussion on this important intersection of theory and application.

---

### Comment · Area_Chair_1HjN · 2023-11-22
**Author-Reviewer Discussion ends soon**

Dear Reviewers and Authors,

The discussion phase ends soon. Please check all the comments, questions, and responses and react appropriately.

Thank you!

Best, AC for Paper #4708

---

### Meta-Review · Area_Chair_1HjN · 2023-12-15

**Metareview:**

The paper concerns a problem of learning from aggregated labels, whose importance is increasing because of privacy concerns. The authors study two classes of loss functions, bag-level and instance-level. They show that the instance-level loss can be seen as a regularized form of the bag-level loss. As both approaches have different bias-variance properties they introduce a novel loss interpolating linearly between them. The new loss is then analyzed in an asymptotic regime. The paper also contains numerical experiments illustrating the theoretical insights.

Reviewers agree that the paper is well-written, with interesting theoretical insights concerning an important problem. The downsize of the paper is the theoretical analysis being limited to a very specific setting (linear models, Gaussian distribution, bag characteristics).

**Justification For Why Not Higher Score:**

The downsize of the paper is the theoretical analysis being limited to a very specific setting (linear models, Gaussian distribution, bag characteristics).

**Justification For Why Not Lower Score:**

All reviewers agree that the paper is above the bar (ratings of 6 from all reviewers).

---

### Decision · Program_Chairs · 2024-01-16

Accept (poster)